# Microglial depletion disrupts normal functional development of adult-born neurons in the olfactory bulb

Jenelle Wallace[1,2,3,4], Julia Lord[3], Lasse Dissing-Olesen[4,5], Beth Stevens[4,5,6,7]*, Venkatesh N Murthy[1,2,3]*

[1]Molecules, Cells, and Organisms Training Program, Harvard University, Cambridge, United States; [2]Center for Brain Science, Harvard University, Cambridge, United States; [3]Department of Molecular and Cellular Biology, Harvard University, Cambridge, United States; [4]FM Kirby Neurobiology Center, Boston Children's Hospital, Boston, United States; [5]Harvard Medical School, Boston, United States; [6]Stanley Center for Psychiatric Research, Broad Institute of MIT and Harvard, Cambridge, United States; [7]Howard Hughes Medical Institute, Boston Children's Hospital, Boston, United States

*For correspondence:
Beth.Stevens@childrens.harvard.edu (BS);
vnmurthy@fas.harvard.edu (VNM)

Competing interests: The authors declare that no competing interests exist.

**Abstract** Microglia play key roles in regulating synapse development and refinement in the developing brain, but it is unknown whether they are similarly involved during adult neurogenesis. By transiently depleting microglia from the healthy adult mouse brain, we show that microglia are necessary for the normal functional development of adult-born granule cells (abGCs) in the olfactory bulb. Microglial depletion reduces the odor responses of developing, but not preexisting GCs in vivo in both awake and anesthetized mice. Microglia preferentially target their motile processes to interact with mushroom spines on abGCs, and when microglia are absent, abGCs develop smaller spines and receive weaker excitatory synaptic inputs. These results suggest that microglia promote the development of excitatory synapses onto developing abGCs, which may impact the function of these cells in the olfactory circuit.

## Introduction

Microglia are critically important for normal brain development in the embryonic and early postnatal stages (*Hammond et al., 2018*). Originally thought to be primarily involved in injury and disease, many recent studies have implicated microglia in diverse neurodevelopmental functions (*Tremblay et al., 2011*; *Salter and Beggs, 2014*; *Wu et al., 2015*; *Hong et al., 2016*). However, much less is known about what role microglia might play in the healthy adult brain, even during the process of adult neurogenesis, which can be thought of as an extension of developmental processes throughout the lifespan.

During early postnatal development, microglia have been implicated in the regulation of synaptic development, including activity-dependent synaptic pruning (*Stevens et al., 2007*; *Schafer et al., 2012*; *Tremblay et al., 2010*; *Paolicelli et al., 2011*; *Gunner et al., 2019*) on one hand and promotion of synaptic development and maturation on the other (*Hoshiko et al., 2012*; *Zhan et al., 2014*; *Miyamoto et al., 2016*; *Nakayama et al., 2018*). Although microglia seem well-positioned to perform similar roles to facilitate the integration of adult-born neurons into circuits in the adult brain in the dentate gyrus (DG) and olfactory bulb (OB) (*Ekdahl, 2012*; *Rodríguez-Iglesias et al., 2019*), most studies on microglial regulation of adult neurogenesis to date have focused on early stages of the process occurring in the neurogenic niches. For example, hippocampal adult neurogenesis is impaired in models of neuroinflammation (*Monje et al., 2003*; *Ekdahl et al., 2003*) and in immune-

**eLife digest** The brain has its own population of resident immune cells known as microglia, which defend against infections and are involved in conditions such as Alzheimer's, Parkinson's and other diseases. In the last decade, new studies have suggested that these cells also sculpt brain circuits during early development. They can 'eat' weak connections between neurons, and help strong ones to mature.

Most of brain 'wiring' happens during development, when the majority of neurons is born and connects together. However, a few brain areas can incorporate new neurons during adulthood into existing circuits. In mice for example, this process takes place in the olfactory bulb, the area that first processes smells: it is believed that new neurons connecting to existing ones helps to detect new odors. It is unclear, however, whether microglia also help to shape these connections, or if their role is confined to early development.

To investigate this question, Wallace et al. gave adult mice a drug that kills only microglia, and then examined how the neurons respond when the animals are exposed to smells. The results show that the new neurons that developed without microglia responded to fewer odors. These neurons also formed weaker connections and had physical features that indicated they might not have been properly incorporated into the circuit.

It may be possible to encourage new neurons to be born in brain areas that normally do not produce these cells in adulthood. Ultimately, this could potentially help to repair the damages of age or disease, but this will rely on understanding exactly how new neurons are integrated into existing brain circuits. Future work, however, is still necessary to figure out how much these new neurons could compensate for cells damaged by injury or disease.

deficient mice (*Ziv et al., 2006*). Microglia regulate adult neurogenesis in the subgranular zone of the DG through ongoing phagocytosis of apoptotic neuroblasts (*Sierra et al., 2010*), although they do not seem to be similarly involved in the subventricular zone (SVZ) or rostral migratory stream (RMS) (*Kyle et al., 2019*; but see *Ribeiro Xavier et al., 2015* for counterevidence).

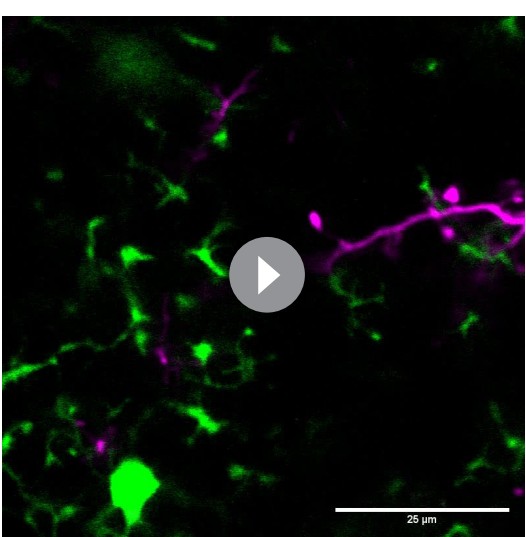

**Video 1.** Z stack showing microglia and abGC dendrites. The movie shows the entire field of view at the beginning of the imaging session for the example shown in *Figure 1*. The z stack volume spans 29 µm and was taken at 32 dpi. The analysis in *Figure 1* was performed on individual planes from such z stacks.
https://elifesciences.org/articles/50531#video1

Most of what is known about microglial involvement in later stages of neurogenesis is based on injury and disease models. Microglial activation via sensory deafferentation in the OB decreases the number of adult-born neurons (*Lazarini et al., 2012*) and their spine density (*Denizet et al., 2017*). In addition, lipopolysaccharide injection or CX3CR1 knockout activates microglia in the hippocampus and alters both inhibitory (*Jakubs et al., 2008*) and excitatory (*Bolós et al., 2018*) synapses onto adult-born neurons in the DG. These studies suggest that microglia can modulate the synaptic integration of adult-born neurons under inflammatory conditions, but raise the question of whether they are similarly involved in the healthy adult brain.

A recent study documented increased activity in principal neurons in the OB after microglial depletion (*Reshef et al., 2017*). Here we investigate the cellular and circuit mechanisms behind this effect by analyzing the functional development of adult-born granule cells (abGCs) in the absence of microglia. We demonstrate that microglial depletion during, but not after abGC development reduces activity in abGCs that make inhibitory connections with principal cells

in the OB. We show that microglia normally interact with spines in developing abGCs, and the volume of these spines is reduced when microglia are ablated. This is accompanied by a reduction in the amplitude of excitatory but not inhibitory inputs to abGCs, suggesting that microglia are essential for proper integration of abGCs in adult circuits.

## Results

### Microglia preferentially interact with spines on abGCs

We labeled cohorts of abGCs born using lentiviral injection into the RMS (*Consiglio et al., 2004*; *Livneh and Mizrahi, 2012*) of adult 8–12 week old mice. To visualize interactions between microglia and abGCs, we performed time-lapse in vivo two-photon imaging of the dendrites of dTomato-labeled abGCs in the external plexiform layer (EPL) of the OB over the first four weeks after injection in CX3CR1-GFP +/- mice, in which microglia are labeled with GFP (*Video 1*, *Figure 1A*). Consistent with previous observations (*Nimmerjahn et al., 2005*; *Tremblay et al., 2010*), we found that microglial processes were highly motile and occasionally appeared in close proximity to labeled dendritic spines (*Video 2*, *Figure 1B*). To quantify whether microglia preferentially interact with dendritic spines (defined as colocalization of a microglial process with at least 5% of the area of a spine head, see Materials and methods, *Analysis of microglia-spine interactions*) on abGCs compared to encountering them by chance during the course of continuous motility, we compared the frequency of interactions between microglial processes and spine heads in the actual imaging data with the frequency of interactions in a series of images in which the microglia channel was arbitrarily shifted with respect to the dendritic imaging channel ('Offsets').

Microglia exhibited an impressive degree of motility, interacting with 38.5% of abGC dendritic spines classified as 'mushroom' spines (*Figure 1C*) and 27.2% of spines classified as 'filopodial' spines (*Figure 1H*) during the course of our 30–90 min imaging sessions, which was not significantly different from the offset data (p=0.13 and p=0.39, respectively) (*Figure 1D,I*). However, we found that microglia interacted significantly more often with individual mushroom spines than predicted by chance (Data: mean 0.15 ± 0.00039 interactions/10 min vs. Offsets: mean 0.12 ± 0.016 interactions/10 min, p=0.048) (*Figure 1E*) though the length of individual interactions was not significantly greater(p=0.96) (*Figure 1F*). In contrast, microglia did not interact with filopodial spines at levels above chance (*Figure 1I–L*). Microglia did not cover significantly more of the spine than predicted by chance during interactions with either spine type (Mushroom: p=0.72, Filopodial: p=0.84) (*Figure 1G,L*).

These results suggest that microglia specifically interact with spines that likely contain functional synapses (*Whitman and Greer, 2007*), positioning them to influence synaptic stabilization and maturation during the early development of abGCs.

### Odor responses are reduced in abGCs that mature in the absence of microglia

To assess whether microglial functions are essential for the development of abGCs, we ablated microglia during the entire time course of abGC development, beginning three weeks before lentiviral labeling (*Figure 2A*). Microglial depletion using the CSF1R inhibitor PLX5622 (hereafter PLX) formulated in chow as previously described (*Elmore et al., 2014*) efficiently ablated microglia from the OB (85% ablation as assessed by immunostaining, 96% ablation as assessed by flow cytometry) within one week and depletion could be maintained at similar levels for up to nine weeks with ongoing delivery (*Figure 2B*, *Figure 2—figure supplements 1* and *2*). We found no evidence of any largescale inflammatory response to microglial depletion, as assessed by immunostaining of glial fibrillary acidic protein (GFAP) (*Figure 2—figure supplement 3*), consistent with previous reports of the effects of PLX on the whole brain (*Elmore et al., 2014*) and the olfactory bulb (*Reshef et al., 2017*).

At five to six weeks post injection, when abGCs have reached a functionally mature state (*Wallace et al., 2017*), we used two-photon imaging to visualize abGC dendrites in vivo. Microglial depletion did not affect the overall number of adult-born neurons in the OB (*Figure 2—figure supplement 4*), consistent with other reports, (*Reshef et al., 2017*; *Kyle et al., 2019*), and we could readily identify dTomato-labeled abGC dendrites in control and PLX-treated mice. Since GCs in the

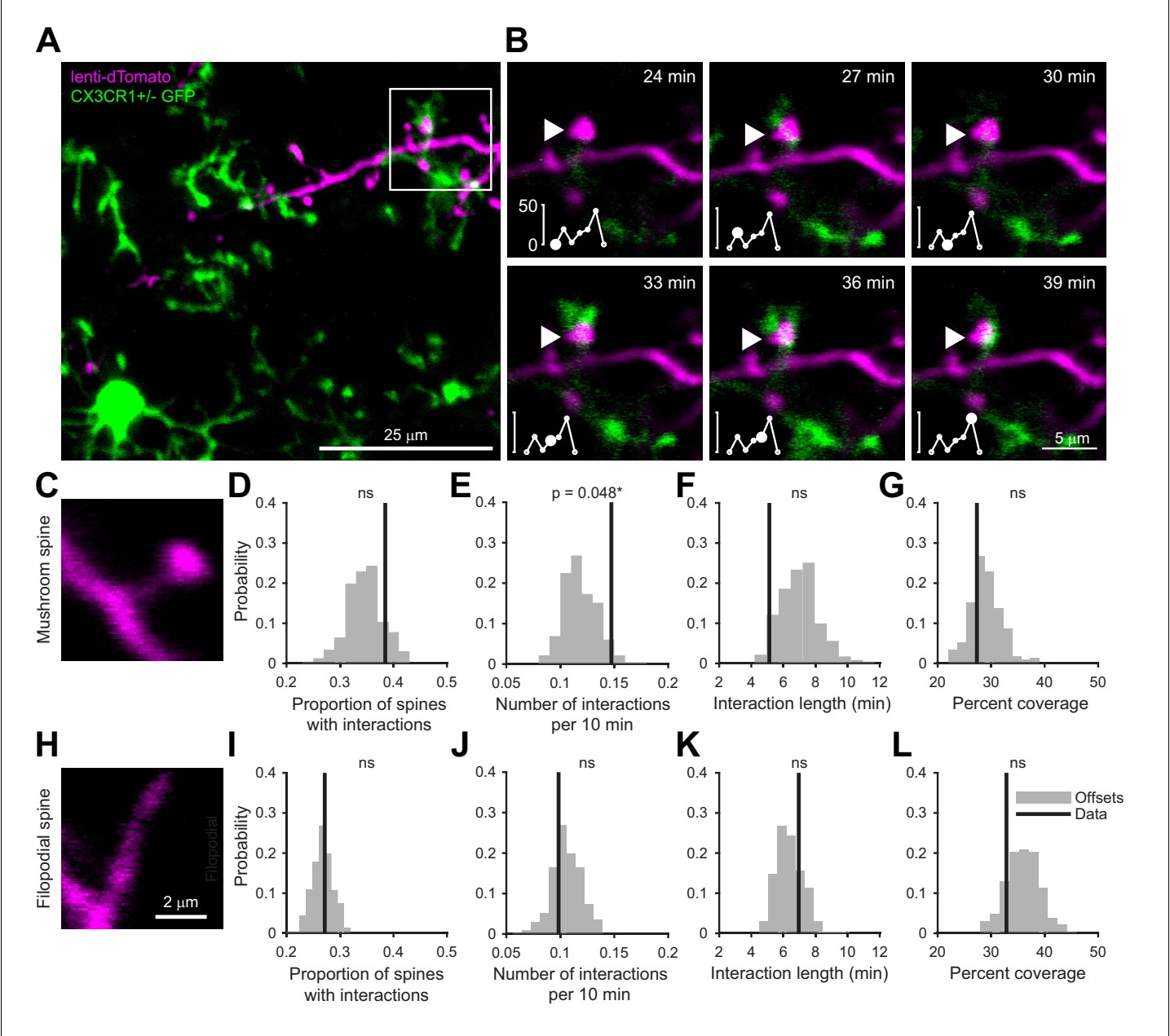

**Figure 1.** Microglia preferentially interact with mushroom spines on developing abGCs. (**A**) Maximum intensity projection (10 μm volume at the first imaging timepoint) showing dTomato-labeled abGCs in a CX3CR1-GFP heterozygous mouse imaged 4 weeks after lentivirus injection. Brightness and contrast adjusted for display only. (**B**) Single plane time series showing the region marked in (**A**). Inset shows the calculated percent microglial coverage for the spine marked with the arrowhead (images shown for six frames, 7th frame not shown but the value is plotted, showing the end of the interaction) with the larger circle marking the value for the corresponding frame. Brightness and contrast adjusted with the same parameters for each timepoint for display only. (**C**) Single plane image showing an example of a spine classified as a mushroom spine because the spine head is wider and brighter than the spine neck. (**D**) Probability distribution showing the proportion of mushroom spines with at least one microglial interaction for mushroom spines in the real data (black line, 'Data') compared to values calculated from iteratively translating the microglial channel relative to the dendritic imaging channel (gray histogram, 'Offsets'). The proportion of spines with interactions was not significantly higher than chance (one-tailed permutation test, p=0.13). (**E**) Probability distribution showing the number of interactions (normalized to 10 min) for mushroom spines. The mean number of interactions (value for each dendrite is the mean number across all mushroom spines on that dendrite) in the real data was significantly higher than chance (one-tailed permutation test, p=0.048). (**F**) Probability distribution showing interaction length for mushroom spines (for spines that had at least one frame that met the criteria for an interaction, see Materials and methods). The mean interaction length across all dendrites (value for each dendrite is the mean interaction length across all interactions for all mushroom spines) was not higher than chance (one-tailed permutation test, p=0.96). (**G**) Probability distribution showing maximum percent coverage (mean across all interactions for a given spine for spines that had at least one frame that

*Figure 1 continued on next page*

*Figure 1 continued*

met the criteria for an interaction). The mean maximum percent coverage across all dendrites (value for each dendrite is the mean interaction length across all interactions for all mushroom spines) was not higher in the real data (one-tailed permutation test, p=0.72). (H) Single plane image of a spine classified as a filopodial spine because it has no clear spine head. (I) Probability distribution showing the proportion of spines with at least one microglial interaction for filopodial spines. The proportion of filopodial spines with interactions was not significantly higher than chance (one-tailed permutation test, p=0.39). (J) Probability distribution showing the number of interactions (normalized to 10 min) for filopodial spines. The mean number of interactions (value for each dendrite is the mean number across all filopodial spines on that dendrite) in the real data was not significantly higher than chance (one-tailed permutation test, p=0.72). (K) Probability distribution showing interaction length for filopodial spines (for spines that had at least one frame that met the criteria for an interaction, see Materials and methods). The mean interaction length across all dendrites (value for each dendrite is the mean interaction length across all interactions for all filopodial spines) was not significantly higher than chance (one-tailed permutation test, p=0.23). (L) Probability distribution showing maximum percent coverage (mean across all interactions for a given spine for spines that had at least one frame that met the criteria for an interaction). The mean maximum percent coverage across all dendrites (value for each dendrite is the mean interaction length across all interactions for all filopodial spines) was not higher in the real data (one-tailed permutation test, p=0.84). n = 726 spines (271 mushroom spines and 455 filopodial spines) from 48 dendrites combined at 1, 2, 3, and 4 weeks post injection in three mice. ns, not significant; *p<0.05.

The online version of this article includes the following source data for figure 1:

**Source data 1.** This spreadsheet contains the values used to create the histograms in *Figure 1D–G and I–L*.

OB are axonless and their release sites are located at dendodendritic synapses on spines in the EPL (*Rall et al., 1966*), we chose to record calcium responses in these dendrites. We first recorded responses in anesthetized mice to a panel of 15 monomolecular odors (Materials and methods, *Odor stimulation*, *Table 1*), while simultaneously imaging morphology in the dTomato channel to aid in region of interest identification and image alignment (*Figure 2C*). AbGC responses to odors were sparse as previously described (*Figure 2D*; *Wallace et al., 2017*), but across the population we could identify dendrites responding to most of the odors in our panel (*Figure 2E*). We characterized responses by taking the mean $\Delta F/F_\sigma$ value over a five-second period following the onset of a two-second odor stimulus and plotted the cumulative distribution of dendritic responses across all odors.

The distribution was shifted left toward lower responsiveness in PLX-treated mice (p=2.56e-08) while the noise distributions constructed from blank trials were not different (p=0.96) (*Figure 2F*). Dendrites in PLX-treated mice also responded to fewer odors (for threshold response criteria, see Materials and methods, In vivo imaging analysis, Thresholds) (median (IQR): Control: 3 (1–6), PLX: 1 (0–4), p=1.16e-04) (*Figure 2G*). We also found that lifetime sparseness (*Willmore and Tolhurst, 2001*), (bounded between 0 and 1, a low score indicates a sparser representation) was lower in dendrites in PLX-treated mice (median (IQR): Control: 0.18 (0.067–0.32), PLX: 0.067 (0–0.25), p=4.18e-04) (*Figure 2H*). These effects were also significant when we performed hierarchical bootstrapping (*Saravanan et al., 2019*) to take into account the fact that we imaged dozens of dendrites from each mouse, with dendrites in PLX-treated mice responding to fewer odors (mean Control: 3.6 ± 0.38, PLX: 2.8 ± 0.26, p=0.050) (*Figure 2—figure supplement 5B*) and having lower median response amplitudes (median Control: 0.14 ± 0.039, PLX: 0.070 ± 0.016, p=0.0011) (*Figure 2—figure supplement 5D*). However, there was no difference in the median amplitude of responses above

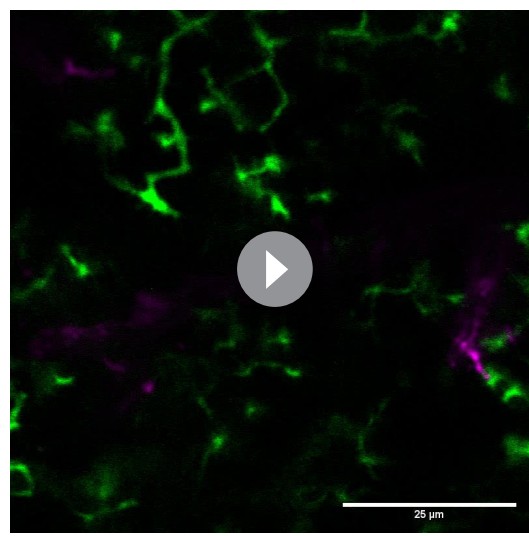

**Video 2.** Time series showing interactions between microglia and dendritic spines of abGCs.
The movie shows a single plane taken from the z stack in *Video 1* across 48 min of imaging (images taken 3 min apart). The time course of the interaction between a microglial process and the mushroom spine shown in the example in *Figure 1* can be observed. The analysis in *Figure 1* was performed on individual planes from such time series.
https://elifesciences.org/articles/50531#video2

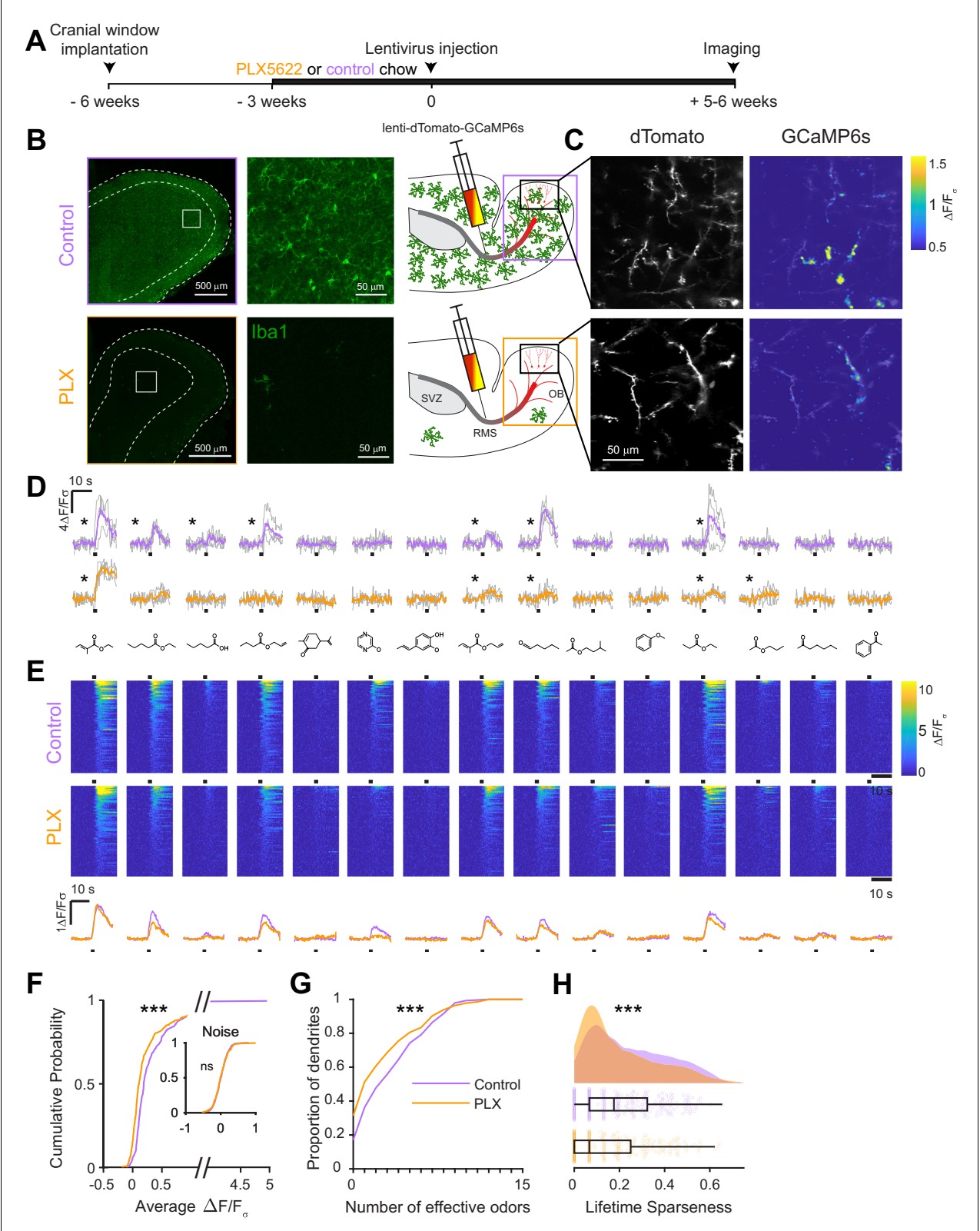

**Figure 2.** Microglial depletion during development reduces odor-evoked responses of abGCs in anesthetized mice. (A) Experimental timeline for microglial depletion during development of abGCs. A cranial window was implanted and 3 weeks later mice were given control or PLX5622-containing chow for the remainder of the experiment. After 3 weeks of chow consumption, a lentivirus was injected into the RMS to label abGCs, which were imaged 5–6 weeks later. (B) Left, images of Iba1 staining in the olfactory bulb of control (top) and PLX-treated (bottom) mice. White squares show the

*Figure 2 continued on next page*

*Figure 2 continued*

locations of the enlarged insets. Dotted lines mark the upper edge of the glomerular and granule cell layers. Right, schematic showing injection of a lentivirus encoding dTomato and GCaMP6s and microglial depletion. (**C**) Example fields of view showing an average intensity projection of dTomato structural images of abGC dendrites (left) and overlaid heatmaps of GCaMP6s-recorded activity (right) in response to ethyl valerate in control (top) and PLX5622-treated (bottom) mice. (**D**) GCaMP6s traces showing odor responses of example ROIs from control (top) and PLX-treated (bottom) mice (chosen to have the same ranked response to the first odor). Gray traces represent responses on individual trials and colored trace is the mean across trials. Individual trial traces were median filtered over three frames before averaging for presentation. *, odor responses for which the mean response was above threshold (**E**) Heatmap traces from the 100 ROIs with the largest odor-evoked Ca2+ signals across all mice ranked separately for each of 15 odors (molecular structures shown above). Black bar denotes odor time. Bottom, mean response time course for each odor across all ROIs. (**F**) Cumulative distribution showing that the distribution of responses (averaged across odors for each dendrite) is shifted to the left in PLX-treated mice (Two sample Kolmogorov–Smirnov test for probability distributions, D = 0.25, p=2.56e-08) while the noise distributions constructed from blank trials are not different (D = 0.042, p=0.96). (**G**) Cumulative distribution showing the number of effective odors (odors that evoked responses above the ROC threshold 0.39, which was calculated across all dendrites from both groups). The median number of effective odors was significantly lower in the PLX-treated group (Wilcoxon rank sum test, z = 3.86, p=1.15e-04). (**H**) Raincloud plot showing the distribution of lifetime sparseness across all dendrites. Above, kernel density estimate. Below, boxplot showing the median, interquartile range (box), and 1.5 times the interquartile range (whiskers) superimposed on a dot plot of all the data (one dot per dendrite). Median lifetime sparseness was significantly lower in the PLX-treated group (Wilcoxon rank sum test, z = 3.53, p=4.18e-04). n = 287 dendrites from five control mice and 277 dendrites from 7 PLX-treated mice. *p<0.05, **p<0.01, ***p<0.001.

The online version of this article includes the following source data and figure supplement(s) for figure 2:

**Source data 1.** This spreadsheet contains values from each dendrite from each mouse for *Figure 2F,G and H*.

**Figure supplement 1.** Quantification of microglial depletion with flow cytometry.

**Figure supplement 2.** Quantification of microglial depletion with immunohistochemistry.

**Figure supplement 3.** Astrocytic response to microglial depletion.

**Figure supplement 4.** Microglial depletion does not affect the number of maturing abGCs.

**Figure supplement 5.** Further analysis of odor responses in abGCs in control versus PLX-treated mice.

threshold (p=0.77) (*Figure 2—figure supplement 5F*), suggesting that the difference in overall responses was mostly mediated by an increase in the proportion of dendrites that did not respond to any of the odors in our panel, which was significantly higher in PLX-treated mice (Control: 0.14 ± 0.039, PLX: 0.32 ± 0.039, p=0.0013) (*Figure 2—figure supplement 5E*).

While imaging in anesthetized mice allows better control of breathing rate, brain motion, and possible motivational influences on brain state, granule cell odor representation is significantly different in awake mice (*Kato et al., 2012*; *Wienisch and Murthy, 2016*; *Wallace et al., 2017*). Therefore, we also imaged abGC dendrites in awake mice and found similar effects as in anesthetized mice (*Figure 3A*, *Figure 3—figure supplement 1*). Dendrites in PLX-treated mice had lower responsiveness (p=0.037) (*Figure 3B*), responding to a lower median number of odors (median (IQR) Control: 1 (0–6), PLX: 0 (0–3), p=0.052) (*Figure 3C*) and having lower lifetime sparseness (median (IQR) Control: 0.067 (0–0.28), PLX: 0 (0–0.16), p=0.056) (*Figure 3D*). Interestingly, while response time courses were similar between control and PLX-treated mice in the anesthetized state (p=0.30), allowing us to characterize responses with a simple mean across the odor analysis period, principal components analysis of the ΔF/F$_\sigma$ traces for all cells' responses to all odors revealed different response time courses in awake mice (p=0.004) (*Figure 3—figure supplement 1*). These differences were likely not due to changes in active sampling of odors since sniffing rates were not different during baseline or odor presentation periods (mean Baseline: Control = 3.49 Hz, PLX = 3.52 Hz, p=0.91; Odor: Control = 4.00 Hz, PLX = 3.94 Hz, p=0.81) (*Figure 3—figure supplement 2*). To ensure that our analysis of response amplitudes was not complicated by this possible change in response timing, we also applied an event detection analysis method (using a sliding window across the response period to detect increases in fluorescence above a noise threshold, see Materials and methods, In vivo Imaging analysis, Event detection) to the awake data and found similar results with dendrites in PLX mice still characterized by responses to a lower median number of odors (p=0.037) (*Figure 3—figure supplement 1*).

## Microglial depletion after abGC development has no effect on odor responses

We next wondered whether the effect of microglial depletion was specific to abGCs developing in the absence of microglia or whether it might affect abGCs more generally. To address this question,

we modified our experimental timeline to label abGCs and wait three months for them to mature fully (*Figure 4A*) before imaging their responses to the same set of odors (*Figure 4B*) in the same mice before and after three weeks of PLX5622. In this case, we found no significant differences in the distribution of responses (p=0.45) (*Figure 4C*), median number of odors evoking a significant response (median (IQR) Control: 1.5 (0–6), PLX: 2 (0–5), p=1.00) (*Figure 4D*), or lifetime sparseness (median (IQR) Control: 0.096 (0–0.31), PLX: 0.099 (0–0.27), p=0.86) (Figure 4E). We verified that our imaging paradigm was stable since there was also no change in responses when we imaged the same mice for two sessions three weeks apart without any PLX treatment (*Figure 4—figure supplement 1*). Even after 9 weeks of PLX treatment, the level of responsiveness remained stable in mature abGCs (*Figure 4—figure supplement 2*).

## Synapse development in abGCs that mature in the absence of microglia

Since we found that microglial depletion reduces the functional responses of abGCs, we wondered if there were accompanying changes in excitatory synapses made on abGCs. We studied spines on the apical dendrites of abGCs in the EPL since our in vivo imaging showed lower calcium responses to odors in these dendrites, which could reflect fewer or weaker synaptic inputs. Four weeks after lentiviral labeling, we examined spines on apical dendrites in abGCs in tissue sections from control and PLX-treated mice (*Figure 5A,B*). We found higher spine density (median (IQR): Control: 0.30 (0.21–0.36) spines/μm, PLX: 0.40 (0.31–0.48) spines/μm, p=0.039) (*Figure 5C*), but smaller spine head volumes in PLX-treated mice (mean Control: $0.37 \pm 0.03$ μm$^3$, PLX: $0.31 \pm 0.02$ μm$^3$, p=0.057) (*Figure 5D*). Furthermore, most of the increase in spine density in PLX-treated mice was due to an increase in the density of filopodial spines that had similar spine head volumes as in control mice (*Figure 5—figure supplement 1*). In contrast, mushroom spines had significantly smaller head volumes in PLX-treated mice (mean Control: $0.45 \pm 0.04$ μm$^3$, PLX: $0.36 \pm 0.03$ μm$^3$, p=0.036) (*Figure 5—figure supplement 1*).

We next investigated the electrophysiological correlates of the observed differences in spine head size by recording spontaneous excitatory postsynaptic currents (sEPSCs) in abGCs. We used the same timeline as our in vivo imaging experiments, namely microglial depletion beginning three weeks before lentiviral labeling, continuing until electrophysiological recordings from labeled cells in brain slices at five to six weeks post injection (*Figure 6A,B*). We found a similar frequency of sEPSCs in cells from control and PLX-treated mice (mean Control: $30.8 \pm 2.2$ Hz, PLX: $30.6 \pm 2.4$ Hz, p=0.48) (*Figure 6C*), but their amplitude was reduced (mean Control: $10.8 \pm 0.6$ pA, PLX: $9.0 \pm 0.4$ pA, p=0.007) (*Figure 6D*). Passive membrane properties including membrane resistance (median Control: 597 MΩ, PLX: 532 MΩ, p=0.31) and capacitance (median Control: 14.2 pF, PLX: 13.6 pF, p=0.61) were unchanged, signifying no differences in cell surface area or resting membrane properties (*Figure 6—figure supplement 1*). We also confirmed that our recording conditions were consistent by verifying that series resistance and the distance of the recorded cells from the mitral cell layer were not different between groups (*Figure 6—figure supplement 1*).

To check whether there might be accompanying changes in inhibition that could offset or augment the observed changes in excitation, we also recorded spontaneous inhibitory postsynaptic currents (sIPSCs) in the same cells (*Figure 6E*). We found no difference in the frequency (p=0.77) (*Figure 6F*) or amplitude of sIPSCs (p=0.79) (*Figure 6G*), suggesting that abGCs that mature in the absence of microglia receive weaker excitatory inputs without noticeable accompanying changes in inhibition.

## Microglial depletion after abGC development has no effect on synaptic inputs

Since there was no significant change in functional responses in abGCs that matured before microglia ablation, we checked whether synaptic inputs were also unchanged in this condition using the same experimental timeline as before and recording sEPSCs in abGCs that experienced three weeks of microglial depletion after three months of maturation (*Figure 7A*). There was no significant change in the frequency (p=0.76) (*Figure 7B*) or amplitude of sEPSCs (p=0.09) (*Figure 7C*). Inhibitory inputs were also unchanged (*Figure 7—figure supplement 1*). These results suggest that microglial depletion only affects synaptic inputs to abGCs when it occurs during the first five to six weeks of the cells' development rather than after maturation.

**Table 1.** Common names and chemical structures for monomolecular odor stimuli.

| | | |
|---|---|---|
| Ethyl tiglate | Valeraldehyde | |
| Ethyl valerate | Isoamyl acetate | |
| Valeric acid | Anisole | |
| Allyl butyrate | Ethyl propionate | |
| Carvone | Propyl acetate | |
| 2-methoxypyrazine | 2-heptanone | |
| Isoeugenol | Acetophenone | |
| Allyl tiglate | | |

## Discussion

We delineate an important role for microglia in the regulation of adult-born neuron integration. Our study adds to the growing literature documenting physiological and/or behavioral effects following microglial depletion from the healthy adult brain (*Parkhurst et al., 2013*; *Torres et al., 2016*; *Reshef et al., 2017*). Importantly, our study is the first to link microglial depletion directly to changes in the in vivo activity and functional inputs of a specific affected cell population. We show that eliminating microglia has functional consequences for abGCs incorporating into the circuitry of the OB, reducing their responses to stimuli. Furthermore, we go on to investigate the physiological mechanisms and show that the dampened responses we observe are consistent with reduced spine volume and weaker excitatory inputs in neurons that develop in the absence of microglia.

### Methodological considerations

Several technical caveats must be acknowledged to allow proper interpretation of our study. While ablation of microglia with PLX5622 is highly efficient, it necessarily involves a period of microglial cell death, and it is unknown how this debris may be cleared or what effects it may have on other cell types. We have attempted to minimize these effects by beginning experiments at least three weeks after initiation of PLX5622 treatment, when the number of microglia should have reached steady state levels (*Elmore et al., 2014*), but we cannot exclude the possibility of long-term effects resulting from microglial cell death. Encouragingly, there seems to be no upregulation of cytokines or GFAP with PLX treatment generally (*Elmore et al., 2014*) or in the OB, specifically (*Figure 2—figure supplement 3*; *Reshef et al., 2017*) in contrast to other methods for microglial depletion, which may cause a significant increase in GFAP+ astrocytes and a cytokine storm (*Bruttger et al., 2015*). However, we acknowledge that the array of possible inflammatory responses to microglial disruption is still poorly understood (*Liddelow and Barres, 2017*) and may involve both responses to microglial cell death as well as reactions to the absence of normal homeostatic cues. Future work will be needed to characterize changes in crosstalk among different cell populations after microglial depletion and implications for the inflammatory environment. However, inflammation in the OB has been shown to generally reduce adult-born neuron numbers (*Lazarini et al., 2012*) and spine density (*Denizet et al., 2017*) as opposed to increasing spine density and reducing excitatory synapse strength as we show here, so it is unlikely that undetected inflammation relating to microglial depletion can fully account for the results we observe.

We also note that microglial depletion with PLX5622 is brain-wide. AbGCs receive local inputs from within the OB as well as feedback from other cortical areas and neuromodulatory inputs

(*Lepousez et al., 2013*), so we cannot unambiguously attribute the effects on abGCs to changes within the OB. Future work will be necessary to investigate whether there is a change in the balance of distal (predominantly feedforward) and proximal (mostly feedback) inputs or instead a more general effect on excitatory synapse maturation and/or maintenance.

## AbGCs in the olfactory circuit

AbGCs become responsive to odor stimuli soon after they arrive in the OB and then undergo a period of functional refinement during which their initially broadly tuned responses become more selective (*Wallace et al., 2017*) (although there may also be a subpopulation of abGCs that shows

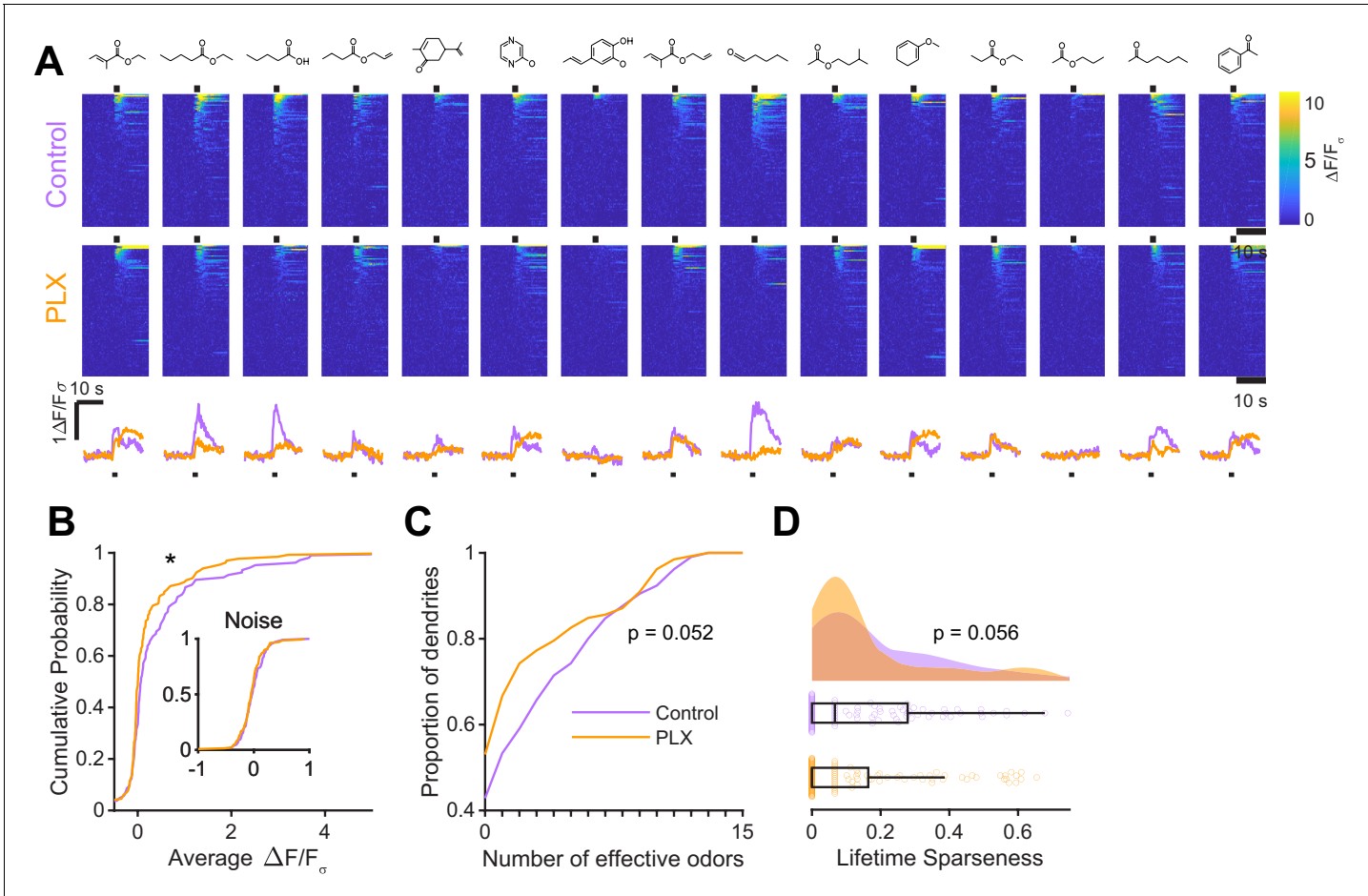

**Figure 3.** Microglial depletion during development reduces odor-evoked responses of abGCs in awake mice. (**A**) Above, Heatmap traces from the 100 ROIs with the largest odor-evoked Ca2+ signals across all mice ranked for each of 15 odors (molecular structures shown above). Below, Mean response time course for each odor across all ROIs. Black bar denotes odor time. (**B**) Cumulative distribution showing that the distribution of responses (averaged across odors for each dendrite) is shifted to the left in PLX-treated mice (Two sample Kolmogorov–Smirnov test for probability distributions, D = 0.18, p=0.037) while the noise distributions constructed from blank trials are not different (D = 0.11, p=0.45). (**C**) Cumulative distribution showing the number of effective odors (odors that evoked responses above the ROC threshold 0.52, which was calculated across all dendrites from both groups). There was a trend toward a lower median number of effective odors in the PLX-treated group (Wilcoxon rank sum test, z = 1.95, p=0.052). (**D**) Raincloud plot showing the distribution of lifetime sparseness across all dendrite. Above, kernel density estimate. Below, boxplot showing the median, interquartile range (box), and 1.5 times the interquartile range (whiskers) superimposed on a dot plot of all the data (one dot per dendrite). There was a trend toward lower median lifetime sparseness in the PLX-treated group (Wilcoxon rank sum test, z = 1.91, p=0.056). n = 105 dendrites from three control mice and 132 dendrites from 4 PLX-treated mice. *p<0.05.

The online version of this article includes the following source data and figure supplement(s) for figure 3:

**Source data 1.** This spreadsheet contains values from each dendrite for *Figure 3B,C and D*.

**Figure supplement 1.** Further analysis of abGC odor responses in awake versus anesthetized mice.

**Figure supplement 2.** Sniffing rates are not different in control versus PLX-treated mice.

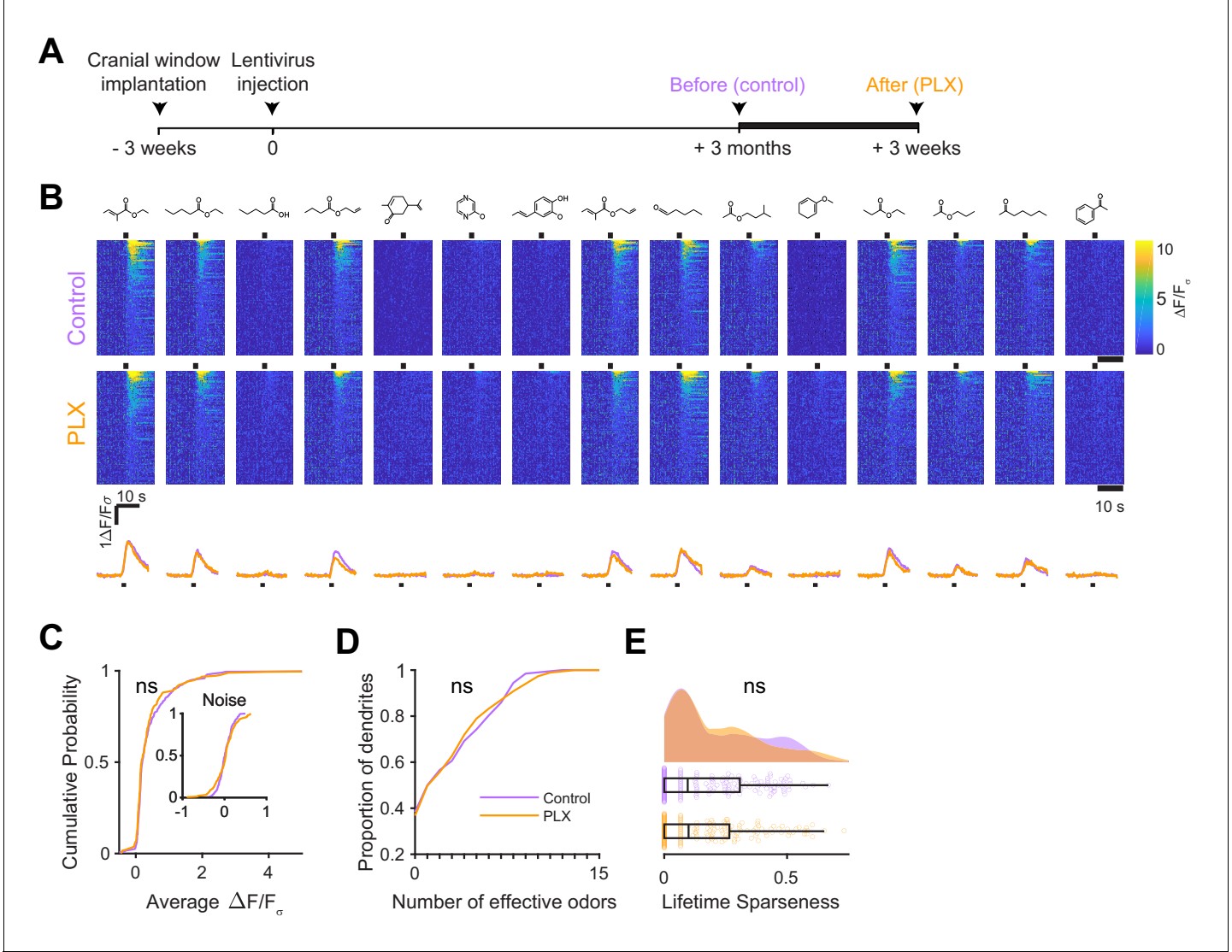

**Figure 4.** Microglial depletion after development has no effect on odor-evoked responses of abGCs. (**A**) Experimental timeline for microglial depletion after development of abGCs. AbGCs were labeled via lentivirus injection and allowed to mature for 3 months. A control imaging session was performed immediately before administration of PLX chow and the second imaging session occurred 3 weeks later. (**B**) Heatmap traces from the 100 ROIs with the largest odor-evoked Ca2+ signals across all mice ranked for each of 15 odors (molecular structures shown above). Black bar denotes odor time. Bottom, mean response time course for each odor across all ROIs. (**C**) Cumulative distribution showing that the distribution of responses (averaged across odors for each dendrite) is not different before and after PLX diet administration (Two sample Kolmogorov–Smirnov test, D = 0.087, p=0.45) and the noise distributions constructed from blank trials are also not different (D = 0.11, p=0.16). (**D**) Cumulative distribution showing the number of effective odors (odors that evoked responses above the ROC threshold 0.53, which was calculated across all dendrites from both groups). The median number of effective odors was not different between groups (Wilcoxon rank sum test, z = 0.0038, p=1.00). (**E**) Raincloud plot showing the distribution of lifetime sparseness across all dendrite. Above, kernel density estimate. Below, boxplot showing the median, interquartile range (box), and 1.5 times the interquartile range (whiskers) superimposed on a dot plot of all the data (one dot per dendrite). Median lifetime sparseness was not different between groups (Wilcoxon rank sum test, z = 0.18, p=0.87). n = 198 dendrites before and 185 dendrites after 3 weeks of PLX administration from three mice. ns, not significant.

The online version of this article includes the following source data and figure supplement(s) for figure 4:

**Source data 1.** This spreadsheet contains values from each dendrite for *Figure 4C,D and E*.
**Figure supplement 1.** Odor responses are stable in mature abGCs.
**Figure supplement 2.** Microglial depletion for a prolonged period after abGC development has no effect on odor-evoked responses of abGCs.

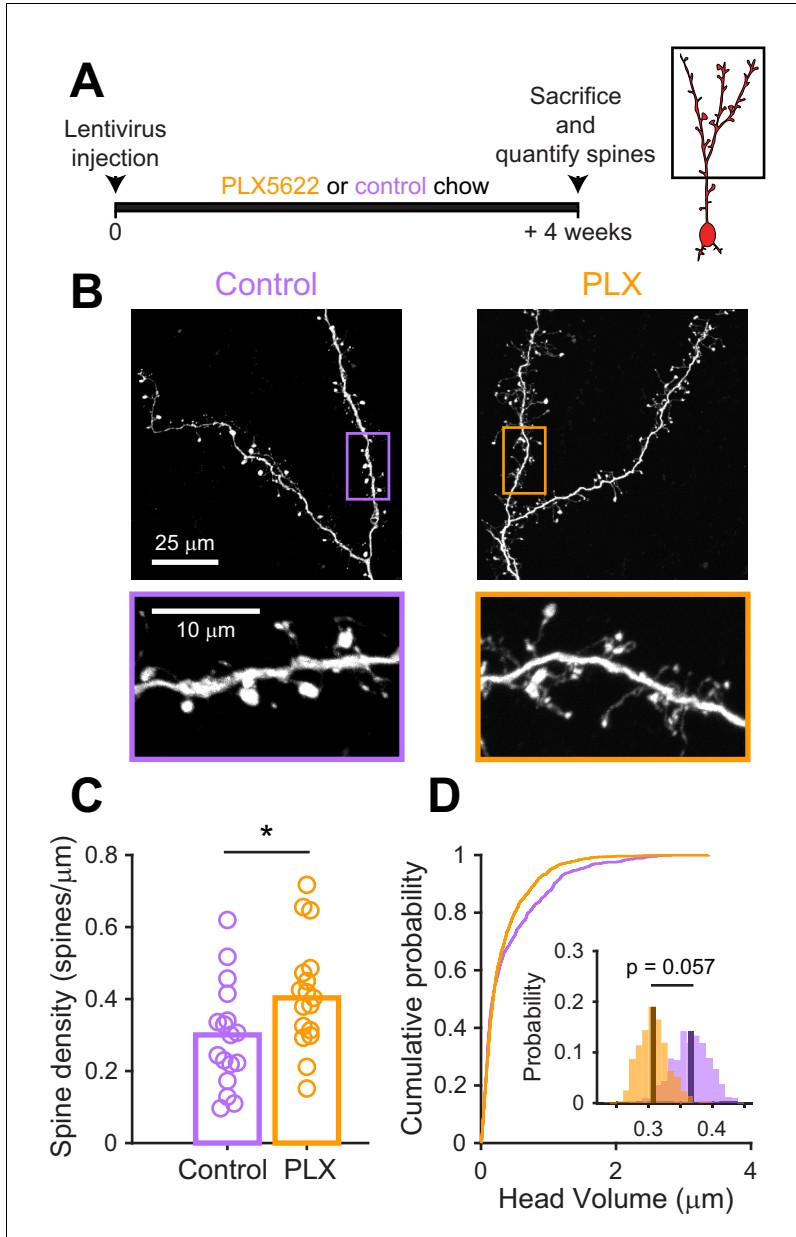

**Figure 5.** Microglial depletion during development reduces spine head volume in abGCs. (**A**) Experimental timeline for microglial depletion during development of abGCs. Mice were given control or PLX5622-containing chow on the same day that a lentivirus was injected into the RMS to label abGCs. Spine numbers and morphology were quantified 4 weeks later. (**B**) Above, sample images showing two apical dendrites from one cell that were analyzed in a control mouse (left) and PLX-treated mouse (right). Below, insets from the images shown above, showing spine morphology in more detail. (**C**) Spine density averaged across 1–5 apical dendrites from each abGC. Spine density was higher in PLX-treated mice (Wilcoxon rank sum test, z = −2.07, p=0.039). Bars indicate medians across cells (circles). (**D**) Cumulative distribution showing the volume of all spines analyzed. Inset, sampling distributions of the mean head volume obtained with the hierarchical bootstrap. The mean head volume (dark lines) was greater in control versus PLX-treated mice (hierarchical bootstrap, p=0.057). n = 810 spines from 17 abGCs from three control mice and 1551 spines from 17 abGCs from 4 PLX mice. *p<0.05.

The online version of this article includes the following source data and figure supplement(s) for figure 5:

**Source data 1.** This spreadsheet contains spine density values for each dendrite for *Figure 5C* and values for the head volume of each spine from each dendrite for *Figure 5D*.

**Figure supplement 1.** Further analysis of spine density and volume in control versus PLX-treated mice.

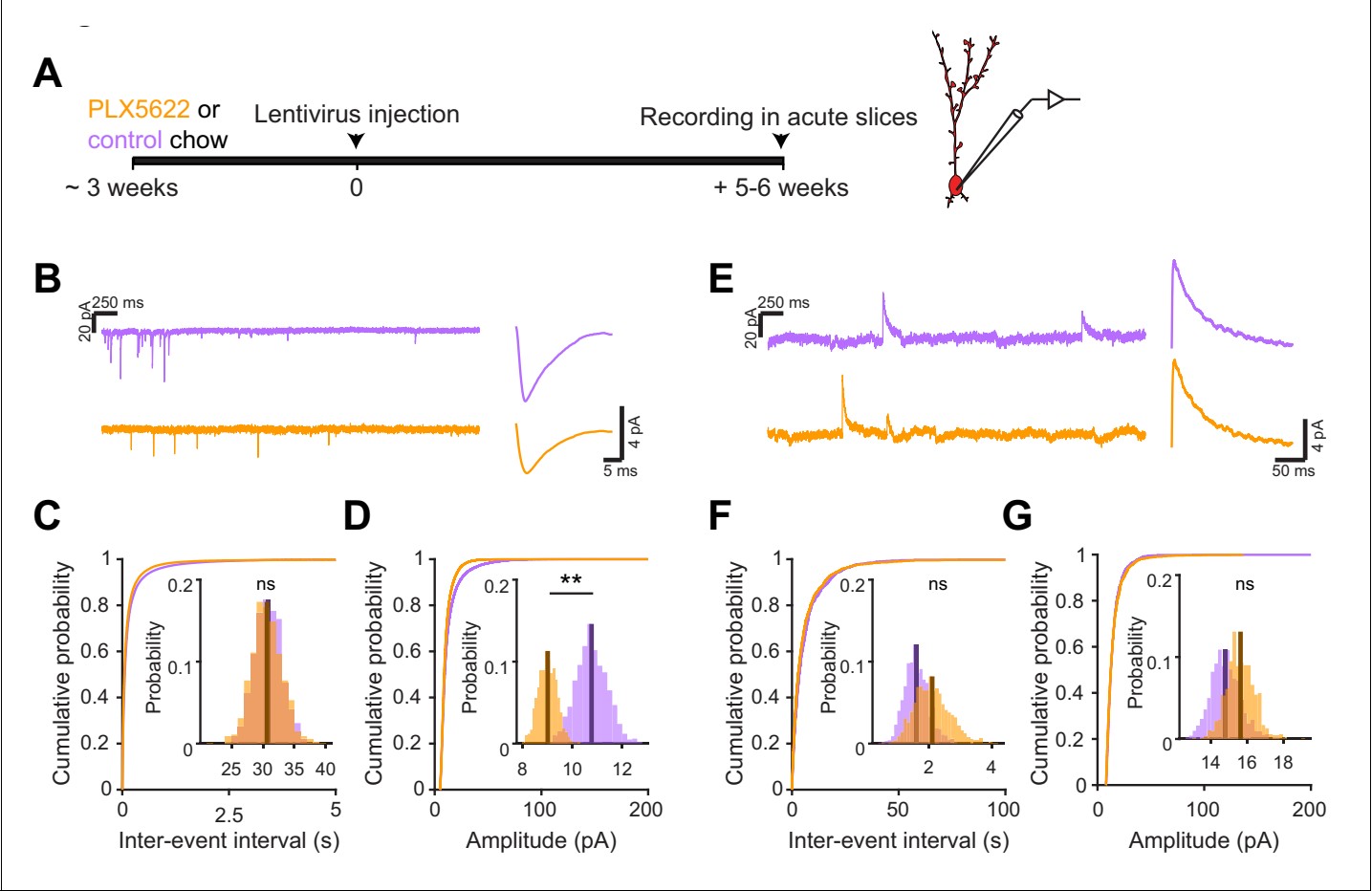

**Figure 6.** Microglial depletion during abGC development reduces the amplitude of excitatory synaptic currents but does not affect inhibitory synaptic currents. (A) Experimental timeline for electrophysiological recording in abGCs (B) Left, sample sections from raw traces recorded from abGCs in control (top) and PLX-treated (bottom) mice. Right, median EPSCs across all EPSCs detected from all mice. (C) Cumulative distribution showing the inter-event intervals from all recorded EPSCs. Inset, sampling distributions of the mean frequency obtained with the hierarchical bootstrap. The median frequencies (dark lines) were not significantly different (hierarchical bootstrap, p=0.48). (D) Cumulative distribution showing the amplitudes from all recorded EPSCs. Inset, sampling distributions of the mean amplitude obtained with the hierarchical bootstrap. The mean amplitude (dark line) was significantly higher in control cells (hierarchical bootstrap, p=0.0068). (E) Left, Sample sections from raw traces recorded from abGCs in control (top) and PLX-treated (bottom) mice. Right, median IPSCs across all IPSCs detected from all mice. (F) Cumulative distribution showing the inter-event intervals from all recorded IPSCs. Inset, sampling distributions of the median frequency obtained with the hierarchical bootstrap. The mean frequencies (dark lines) were not significantly different (hierarchical bootstrap, p=0.77). (G) Cumulative distribution showing the amplitudes from all recorded IPSCs. Inset, sampling distributions of the mean amplitude obtained with the hierarchical bootstrap. The mean amplitudes (dark lines) were not significantly different (hierarchical bootstrap, p=0.79). For EPSCs: n = 30 abGCs from four control mice and 33 abGCs from 4 PLX mice. For IPSCs: n = 29 abGCs from four control mice and 30 abGCs from 4 PLX mice (same mice in both cases and cells used for both EPSCs and IPSCs if the recordings met criteria stated in Materials and methods). ns, not significant; **p<0.01.

The online version of this article includes the following source data and figure supplement(s) for figure 6:

**Source data 1.** This spreadsheet contains the frequency and amplitude values for all detected events from each dendrite for *Figure 6C,D,F and G*.
**Figure supplement 1.** Passive electrophysiological properties and recording conditions are similar for abGCs in control versus PLX-treated mice.

broadening of responses with maturation *Quast et al., 2017*; *Wallace et al., 2017*). Although the mechanism behind this increase in stimulus selectivity is not well-understood, it may involve both a decrease in some aspects of excitability (*Carleton et al., 2003*; *Nissant et al., 2009*), especially dendritic excitability (*Wallace et al., 2017*), as well as selective reorganization of synaptic inputs such that mature abGCs become more responsive to a particular glomerular module at the expense of other inputs. We propose that abGCs in PLX-treated mice experience a normal drop in excitability (likely regulated by cell-intrinsic mechanisms) without a concomitant selective strengthening of

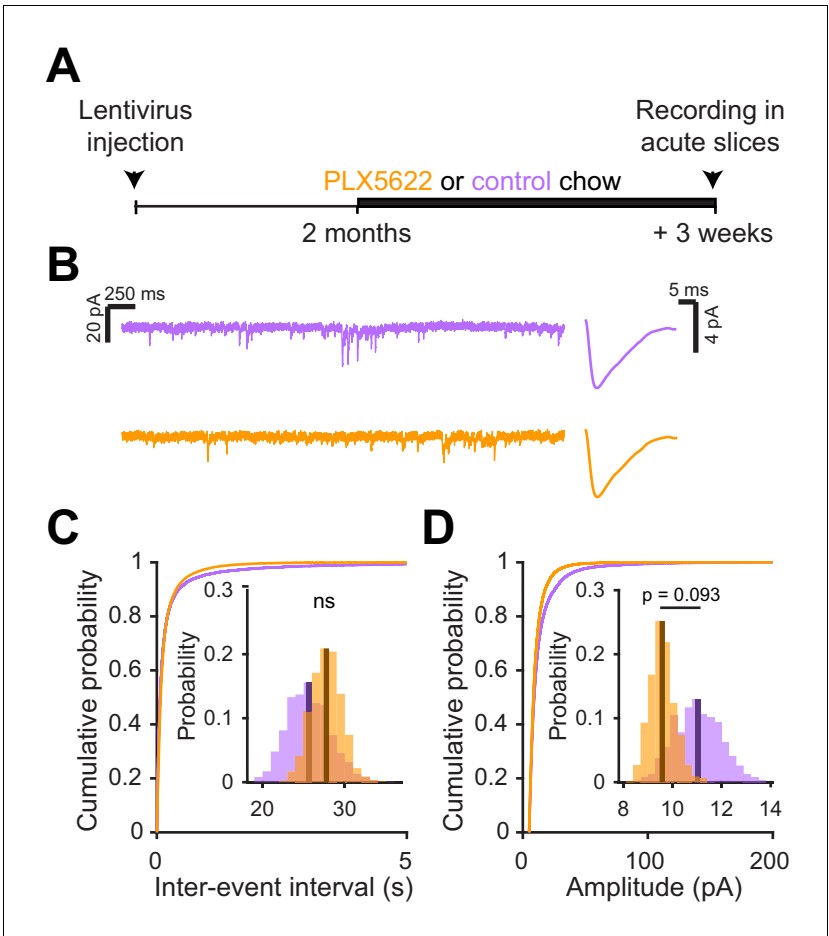

**Figure 7.** Microglial depletion after abGC development has no effect on excitatory synaptic currents. (**A**) Experimental timeline for electrophysiological recordings in abGCs after their development (**B**) Left, Sample sections from raw traces recorded from abGCs in control (top) and PLX-treated (bottom) mice. Right, median EPSCs detected from all mice. (**C**) Cumulative distribution showing the inter-event intervals from all recorded EPSCs. Inset, sampling distributions of the mean frequency obtained with the hierarchical bootstrap. The mean frequencies (dark lines) were not significantly different (hierarchical bootstrap, p=0.76). (**D**) Cumulative distribution showing the amplitude from all recorded EPSCs. Inset, sampling distributions of the mean frequency obtained with the hierarchical bootstrap. The mean amplitudes (dark lines) were not significantly different (hierarchical bootstrap, p=0.093). n = 23 abGCs from three control mice and 27 abGCs from 3 PLX mice. ns, not significant. The online version of this article includes the following source data and figure supplement(s) for figure 7:

**Source data 1.** This spreadsheet contains the frequency and amplitude values for all sEPSCs from each dendrite for *Figure 7C and D*.

**Figure supplement 1.** Microglial depletion after abGC development has no effect on inhibitory synaptic currents.

specific synaptic inputs, leading to sparser odor responses (*Figure 8*). This is consistent with the higher density of more filopodial-like spines in PLX-treated mice (characteristic of abGCs at an earlier stage of maturation *Breton-Provencher et al., 2014*) along with smaller mushroom spines (*Figure 5*) and lower amplitude sEPSCs (*Figure 6*). It is also possible that abGCs in PLX-treated mice could have generally lower excitability, but we regard this as unlikely since excitability has been mainly shown to affect survival rather than synaptic inputs in abGCs (*Lin et al., 2010*), in contrast to what we observe here. Future work will be needed to disentangle the different aspects of abGC maturation and more fully elucidate which are affected by microglia.

Though the effects on synaptic development that we observe in individual cells are modest (~15% reduction in median sEPSC amplitude in PLX-treated mice, *Figure 6C*), these effects combined nonlinearly across many synapses likely become important near response threshold in vivo. Indeed, we

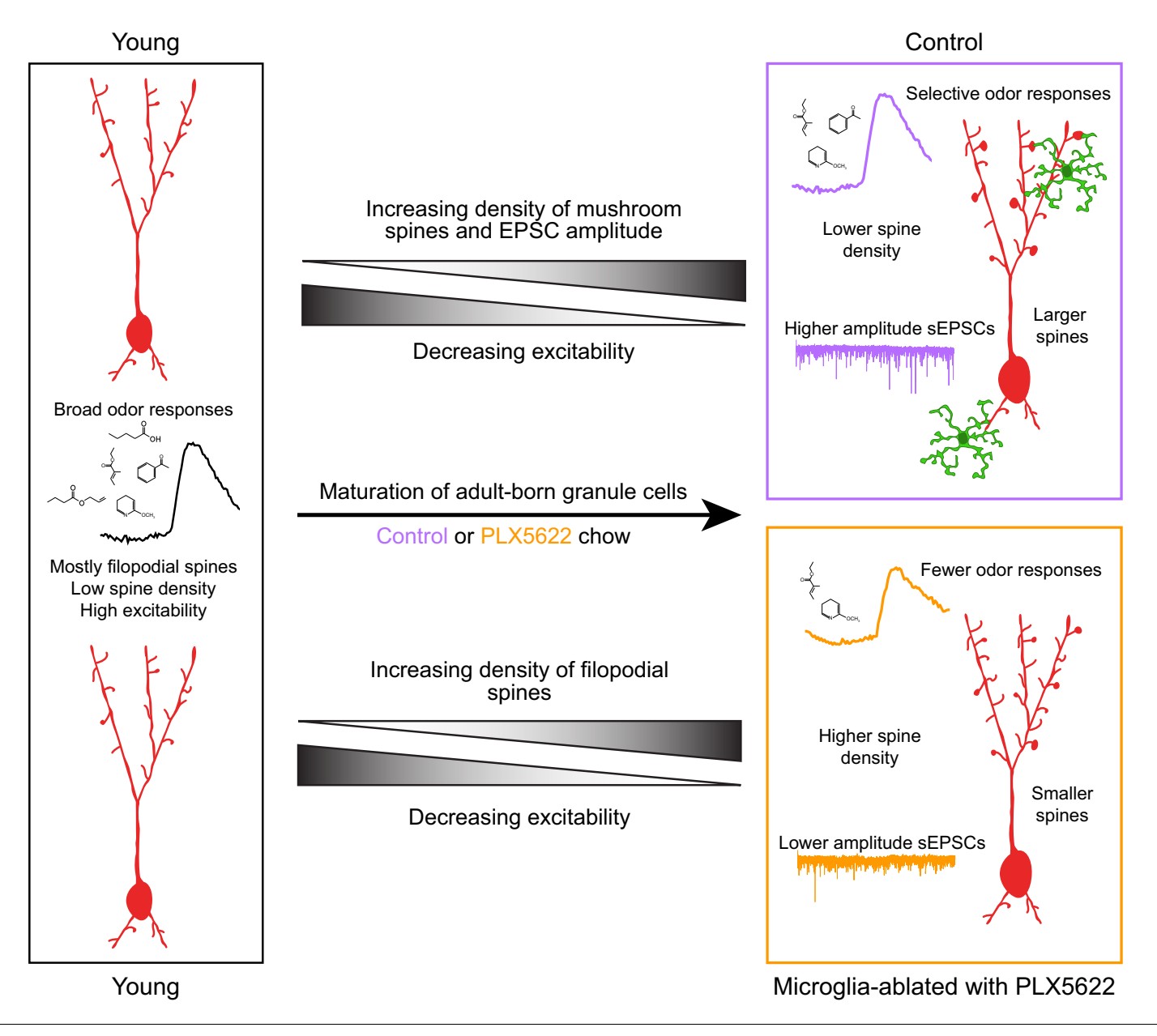

**Figure 8.** Summary of results and model for the role of microglia in abGC maturation. Young abGCs enter existing circuits and begin to make synapses, a process that involves the extension of filopodial spines that sample potential synaptic partners (*Breton-Provencher et al., 2014*). In control mice, abGCs undergo a coordinated process of synaptic formation and elimination (*Mizrahi, 2007*; *Sailor et al., 2016*), which leads to an increasing number of mushroom spines with mature synapses and higher frequency and amplitude excitatory synaptic currents as they mature (*Whitman and Greer, 2007*; *Kelsch et al., 2008*; *Breton-Provencher et al., 2014*). Maturation likely also involves a decrease in dendritic excitability (*Carleton et al., 2003*; *Nissant et al., 2009*; *Wallace et al., 2017*), such that stronger, coordinated synaptic inputs with particular odor tuning are necessary to activate mature abGCs, leading to more selective odor responses. In mice treated with PLX5622 to ablate microglia throughout the time course of abGC development, we hypothesize that abGCs undergo some parts of the maturation process, but not others. We show that aspects of excitability, such as input resistance, mature normally, and overall spine density increases to levels even above controls. However, synapses fail to mature, leading to overall smaller spines and lower amplitude sEPSCs. This may contribute to lower odor responsiveness in abGCs that have matured in the absence of microglia.

observed larger effects on calcium responses recorded in vivo (median response amplitudes halved across all abGCs imaged in each PLX-treated animal, *Figure 2—figure supplement 5D*), which likely represent dendritic calcium spikes or global calcium events accompanying somatic action potentials that result from simultaneous activation of many feedforward and feedback synapses (*Egger et al., 2005*; *Egger, 2008*).

## Microglia-neuron interactions in the adult brain

Though several studies have examined microglial motility (*Nimmerjahn et al., 2005*) and interactions between microglia and neuronal elements (*Wake et al., 2009*; *Tremblay et al., 2010*; *Sipe et al., 2016*), it has been unclear whether microglia interact with synapses more often than would be expected by chance, given the dense synaptic milieu of the adult brain and the high degree of microglial process motility. Using automated methods to segment microglial processes from in vivo two-photon time-lapse imaging experiments and shuffle the resulting images for comparison, we demonstrated targeted motility towards mushroom spines on abGCs, which are likely to contain established excitatory synapses (*Whitman and Greer, 2007*). In the absence of microglia, we suggest that fewer spines attain mushroom morphology, leading to overall reduced spine volume and correspondingly weaker excitatory synapses. Concomitantly, we observed an increase in spine density in microglia-ablated animals, which is likely accounted for by filopodial spines which did not have functional or stable synapses (since we did not see an accompanying increase in the frequency of synaptic currents in PLX-treated mice). One potential unifying hypothesis is that microglia may prune weaker synapses, allowing stronger synapses to strengthen further (*Stevens et al., 2007*; *Schafer et al., 2012*; *Tremblay et al., 2010*; *Paolicelli et al., 2011*). However, unlike other sensory systems that experience largescale pruning events during development, abGCs develop an increasing number of synapses over time (*Carleton et al., 2003*; *Breton-Provencher et al., 2014*). This does not preclude a role for microglia in pruning inputs to these cells (in a situation in which synaptic formation occurs at a higher rate than elimination) but suggests that if this occurs, it is likely to be more subtle and difficult to detect with conventional methods. Although we did not directly observe pruning of dendritic spines in abGCs in our time lapse imaging experiments, it remains possible that microglia could prune presynaptic elements (*Schafer et al., 2012*; *Gunner et al., 2019*) (which we did not image) or that our frame rate was too slow to observe such events (*Weinhard et al., 2018*).

In addition to synaptic pruning, microglia may be involved in promoting synaptic strengthening. Deficits in microglial CX3CR1 have been linked to weaker synapses and impaired connectivity (*Zhan et al., 2014*; *Reshef et al., 2017*) as well as delays in maturation of the AMPA/NMDA ratio (*Hoshiko et al., 2012*). It is still unclear whether these effects, similar to those that we observe in the PLX ablation model, are related to differences in spine surveillance and direct microglial-neuron signaling or due to altered microglial behavior, including changes in release of moderating factors or interactions with other cell types, such as astrocytes. We expect that in the future, a new generation of tools for more specifically modifying microglial gene expression and function during defined time windows will help clarify the mechanisms by which microglia affect synaptic development.

In addition, although most previous work has focused on microglial regulation of synapses, microglia may also regulate neuronal excitability through other mechanisms (*Li et al., 2012*; *Peng et al., 2019*). Accordingly, although we found no differences in passive membrane properties between abGCs in control versus PLX-treated animals, we cannot exclude additional possible effects of microglia on nonsynaptic aspects of abGC development, including active dendritic properties, that could also contribute to the reduced responsiveness we observe when microglia are ablated.

## Timing of microglial depletion

While our results suggest that the effect of microglial depletion is specific to developing abGCs (since our results with microglial depletion after abGC development did not reach statistical significance in most cases), we saw similar trends in reduced responses and excitatory inputs regardless of the timing of microglial depletion. This could be because lentiviral labeling is an imperfect method for isolating a single cohort of abGCs (so some abGCs in the 'after development group' may still be at an earlier stage of development – see discussion of newcomer cells in *Wallace et al., 2017*). Another possibility is that microglial depletion also affects mature abGCs, although to a lesser extent

than developing abGCs. This could be because microglia may be involved in the general maturation or strengthening of newly formed excitatory synapses in GCs, and even mature GCs have high rates of synapse formation (and elimination) compared to cells in other brain regions (*Mizrahi, 2007*; *Sailor et al., 2016*). In this scheme, developing abGCs would demonstrate the most significant phenotype due to their higher rates of spine dynamics, but even mature GCs might accrue smaller effects over time.

The larger effect of microglial depletion on developing rather than mature adult-born neurons highlights the role of microglia during developmental stages and may explain why some studies have not found significant effects of microglial depletion in other areas of the healthy adult brain (*Elmore et al., 2014*; *Torres et al., 2016*).

### Consequences for olfactory processing

The reduced activity that we observe in mature abGCs after microglial depletion is consistent with increased activity in the principal cells they inhibit (*Reshef et al., 2017*). Since reduced inhibition from GCs to OB principal neurons has been directly linked to an increase in the time needed to discriminate odors in challenging olfactory tasks (*Abraham et al., 2010*), this alteration in the OB circuitry could have functional consequences (*Egger and Urban, 2006*). Furthermore, given that abGCs may have an outsized role in the plasticity that underlies complex olfactory behaviors (*Breton-Provencher et al., 2009*; *Jung et al., 2000*; *Mandairon et al., 2018*), microglial regulation of their development may contribute to ongoing plasticity in the olfactory system.

## Materials and methods

**Key resources table**

| Reagent type (species) or resource | Designation | Source or reference | Identifiers | Additional information |
|---|---|---|---|---|
| Genetic reagent (*M. musculus*) | CX3CR1-GFP heterozygote | *Jung et al., 2000* | Cat# Jax: 008451 | |
| Genetic reagent (lentivirus) | pLenti-TRE-t2A-dTomato-GCaMP5 | *Wallace et al., 2017* | | Produced in house (see Materials and methods, 'Viral Vectors') |
| Genetic reagent (lentivirus) | pLenti-TRE-t2A-dTomato-GCaMP6s | *Wallace et al., 2017* | | Produced in house (see Materials and methods, 'Viral Vectors') |
| Genetic reagent (lentivirus) | pLenti-TRE-t2A-dTomato | *Wallace et al., 2017* | | Boston Children's Hospital Viral Core |
| Genetic reagent (lentivirus) | pLenti-hSynapsin - tTad | *Hioki et al., 2009* | | Produced in house or by Boston Children's Hospital Viral Core |
| Antibody | Anti-Iba1 (Rabbit polyclonal) | Wako | Cat # 019–19741 RRID:AB_839504 | 1:500 |
| Antibody | Anti-GFAP (Rabbit polyclonal) | Dako | Cat # Z0334 RRID:AB_10013382 | 1:1000 |
| Antibody | Anti-BrdU (Rat monoclonal) | Abcam | Cat # ab6326 | 1:200 |
| Antibody | Anti-NeuN (Mouse monoclonal) | Millipore | MAB377 | 1:200 |
| Antibody | Anti-CD45 (Rat monoclonal) | Biolegend | 103116 | 1:100 (final concentration of 2 µg/mL) |
| Antibody | Anti-CD11b (Rat monoclonal) | Biolegend | 101217 | 1:200 (final concentration of 2.5 µg/mL) |
| Antibody | Anti CD16/CD32 (Rat) | BD Bioscience | 553141 | 1:50 (final concentration of 10 µg/mL) |

*Continued on next page*

*Continued*

| Reagent type (species) or resource | Designation | Source or reference | Identifiers | Additional information |
|---|---|---|---|---|
| Chemical compound, drug | PLX5622 | Plexxikon | PLX5622 | 1200 mg/kg |
| Chemical compound, drug | BrdU | Sigma Aldrich | Cat # 19–160 | 100 mg/kg |
| Software, algorithm | FlowJo | FlowJo | Version 10.2 | https://www.flowjo.com/solutions/flowjo/downloads/previous-versions |
| Software, algorithm | Imaris | Oxford Instruments | Versions 8 and 9 | https://imaris.oxinst.com/ |
| Software, algorithm | pClamp | Molecular Devices | Version 10.3 | |
| Software, algorithm | Matlab | Mathworks | Version 2019b | |
| Software, algorithm | MiniAnalysis | Synaptosoft | Version 6.0.3 | http://www.synaptosoft.com/MiniAnalysis/ |
| Other | Blue Dead Cell Stain | ThermoFisher | L34961 | |
| Other | Counting Beads (CountBright) | ThermoFisher | C36950 | |

## Mice

Mice were C57BL/6J males (Jackson Laboratories) or CX3CR1-GFP heterozygotes (Jackson Laboratories stock #008451) (*Jung et al., 2000*) that were 8 to 12 weeks-old at the beginning of the experiment. Mice were singly housed after chronic cranial window implantation or housed with littermates for experiments that did not require an implant. In both cases, they were housed on a 12 hr reversed light/dark cycle after window implantation or viral injection. Littermates were randomly assigned to experimental groups (control vs. PLX). All procedures were performed using approved protocols in accordance with institutional (Harvard University Institutional Animal Care and Use Committee) and national guidelines.

## Viral vectors

For all experiments involving lentiviral labeling of abGCs, we used a Tet-Off lentiviral system (*Hioki et al., 2009*) with one construct expressing the transactivator tTAad under control of a synapsin promotor (lenti-STB) and a second construct expressing structural and/or activity markers. For spine quantification in fixed tissue (*Figure 5*) and electrophysiology (*Figures 6* and *7*), we used lenti-STB and lenti-dTomato produced by the Boston Children's hospital. For imaging microglia-spine interactions (*Figure 1*), we used lenti-STB and lenti-dTomato-t2A-GCaMP5 and for all experiments measuring odor responses (*Figures 2*, *3* and *4*) we used lenti-STB and lenti-dTomato-t2A-GCaMP6s produced in house. VSV-G pseudotyped lentiviral vectors were produced by transfection of human embryonic kidney cells (HEK293FT) with third-generation lentivirus plasmids using lipofection (Mirus TransIT−293). Supernatant was collected 48 hr after transfection and concentrated using ultrafiltration (Centricon Plus-20 PLGC centrifuge filter units).

## Lentiviral injections and cranial window surgeries

Reproduced from *Wallace et al. (2017)*:

"Mice were anesthetized with an intraperitoneal injection of ketamine (100 mg/kg) and xylazine (10 mg/kg) and body temperature was maintained at 37°C by a heating pad. A small craniotomy was made bilaterally over the RMS injection sites (coordinates from bregma: A +3.3, L +0.82, from the brain surface: V-2.9 and −2.7) and 250 nL of lentivirus (1:1 mixture of both constructs or 1:50 dilution of the tTA-containing construct to achieve sparser labeling for spine quantification in *Figure 5*) was injected at each of the two depths using a pulled glass micropipette (tip diameter approximately 10–20 um).' For cranial windows, the surface of the brain was kept moist with artificial cerebrospinal fluid (125mMNaCl, 5mMKCl, 10mMGlucose, 10mMHEPES, 2 mM CaCl2 and 2 mM MgSO4 [pH 7.4]) and Gelfoam (Patterson Veterinary) and a glass window consisting of two 3 mm No. one coverslips (Warner) glued together with optical glue (Norland Optical Adhesive 61) was implanted as previously described (*Adam and Mizrahi, 2011*). For mice used for awake imaging, Kwik Sil (World Precision

Instruments) was placed between the coverslip and the brain surface to reduce movement. In this case, the coverslip consisted of two 3 mm and one 4 mm No. 0 coverslips forming a plug (**Dombeck and Tank, 2014**) with the 4 mm coverslip cut with a diamond knife to fit between the mouse's eyes. In both cases, the edges around the coverslip were sealed with Vetbond (3M) and then C and B-Metabond dental cement (Parkell, Inc). A custom-made titanium headplate (eMachine-Shop) was cemented to the skull. After surgery, mice were treated with carprofen (6 mg/kg) every 24 hr and buprenorphine (0.1 mg/kg) every 12 hr for 5 days."

## Two-photon imaging of microglia-spine interactions (*Figure 1*)

A custom-built two-photon microscope (**Wienisch et al., 2011**) was used for in vivo imaging. Fluorophores were excited and imaged with a water immersion objective (20x, 0.95 NA, Olympus) at 950 nm using a Ti:Sapphire laser (Mai Tai HP, Spectra-Physics). The point spread function of the microscope was measured to be 0.66 $\times$ 0.66$\times$2.26 µm. Image acquisition and scanning were controlled by custom-written software in Labview. Emitted light was routed through two dichroic mirrors (680dcxr, Chroma and FF555- Di02, Semrock) and collected by two photomultiplier tubes (R3896, Hamamatsu) using filters in the 500–550 nm range (green channel, FF01-525/50, Semrock) and 572–642 nm range (red channel, FF01-607/70, Semrock). Fields of view were 75 $\times$ 75 µm square spanning 800 $\times$ 800 pixels. Z-stacks of approximately 30 µm depth with a 1 µm z step for both channels (16 bit) were taken every 3 min (0.5 Hz frame rate with 3x averaging during acquisition) for periods of 30–90 min. Two or three fields of view were imaged in each mouse.

## Analysis of microglia-spine interactions (*Figure 1*)

Since both channels exhibited bleed-through with our imaging parameters, the ImageJ spectral unmixing plugin (Author: J. Walter) was used to calculate and apply unmixing matrices for each image stack prior to further analysis. In Fiji, spine heads were delineated manually for each time point in the frame where they appeared brightest using the oval or polygon tools and ROI Manager and classified as either mushroom (spines whose spine head was wider than the spine neck at all timepoints) or filopodial (without a well-defined head). The Weka segmentation plugin (**Arganda-Carreras et al., 2017**) was used to perform binary segmentation of microglial processes from background after training on five frames (that were fully segmented manually) selected to represent a variety of microglial morphologies and brightness variation across the three mice. To optimize our resolution, we segmented very conservatively by using z stacks to mark microglial processes only in the plane where they appeared brightest. This strategy combined with only delineating spine heads in the brightest frame means that we only detected the closest interactions between microglial processes and spine heads. The features chosen for segmentation training in Weka were Gaussian blur, Sobel filter, Hessian, and Difference of Gaussians. This approach allowed us to segment complex microglia morphology from background automatically in every image frame and obviated the need for corrections for bleaching or variations in brightness across different imaging fields. Imaging frames that were too dim to segment (usually due to loss of immersion water) were excluded. Each segmented image stack was checked manually to ensure that any residual bleed-through from the red channel did not appear in the segmentation. After segmentation, ROIs representing spine heads were exported to Matlab using a custom-written macro with the command getSelectionCoordinates. In Matlab, the ROIs were loaded onto the segmented image and the mean value of the binary microglia channel within each ROI at each timepoint (0 if there was no colocalization or up to one if all pixels were colocalized with a segmented microglial process) was measured. The frame was quantified as containing a microglia-spine interaction if the value of colocalization was greater than 0.05 (at least 5% of the spine head area overlapped by a microglial process). To compare the overlap between microglial processes and spine heads in the real data compared to what might be expected by chance, we iteratively translated the microglia channel relative to the marked spine head ROIs to calculate distributions for all the measured interaction parameters (**Dunn et al., 2011**). To do this, we took the original segmented image stack and translated it horizontally, vertically, or horizontally and vertically first by the maximum spine width across the whole data set (~32 pixels) to ensure none of the offsets would overlap the real data and then iteratively by the mean spine width (~10 pixels) to ensure each offset would be as uncorrelated as possible for a total of 231 offsets. Note that this method likely underestimates the significance of interactions in the real dataset because microglia

cell bodies were never colocalized with spines in the real dataset (dendrites overlapping microglia cell bodies were not chosen for imaging) but were likely colocalized and quantified as interacting with spines in some of the offset datasets. To produce *Video 2*, frames were first registered with the MultiStackReg plugin (Author: Brad Busse) based on the magenta channel and bleaching was corrected with histogram matching in ImageJ (these steps were not necessary for analysis because we segmented each image frame separately as described above).

## Two-photon imaging of odor-evoked responses (*Figures 2* and *4*)

Animals were matched in littermate pairs before cranial window surgery. All animals with a clear region of the cranial window and visible lentiviral expression were used for imaging (2/4 control mice and 2/4 PLX-treated mice in the first experiment, 1/2 control mice and 1/2 PLX mice in a second experiment, and 2/4 control mice and 4/4 PLX-treated mice in a third experiment). Imaging was performed at 930 nm with the same two-photon microscope described above.

Reproduced from our previous work (*Wallace et al., 2017*):

"Animals were anesthetized with an intraperitoneal injection of ketamine and xylazine (90% of dose used for surgery) and body temperature was maintained at 37°C by a heating pad. Frame rates were 4 Hz, the pixel size was 0.5 μm, and fields of view measured 150 × 150 μm. To locate regions for imaging, a low magnification z stack (~300–500 μm square) at slow scanning speed (usually 0.5 Hz) with a 1–2 μm z step was taken from the surface of the dura to the granule cell layer. Planes with many dendrites perpendicular to the imaging axis were chosen for imaging during odor stimulation."

## Two-photon imaging in awake animals (*Figure 3*)

A subset of the animals that were imaged under anesthesia were chosen for awake imaging (all animals that had sufficiently stable cranial windows were chosen from each of the experiments). Animals were water-restricted beginning 1–2 days before being handled and accustomed to head-fixation in a restraining tube (*Guo et al., 2014*) for 1–2 days with manual delivery of water rewards (approximately 30 min sessions each). They were then acclimated to the sound of the scan mirrors and odor delivery (using the full odor set) on the day before imaging with manual delivery of water rewards before imaging and periodically between sets of repetitions. The same protocol was repeated for 1 or 2 days of imaging for each mouse.

## Odor stimulation

Odor lists are found in *Table 1*. Odors (Sigma) were delivered with a custom-built 16 channel olfactometer at a nominal volumetric concentration of 16% (v/v) in mineral oil and further diluted by 16 times in air to a final concentration of approximately 1% (except for isoeugenol which was not diluted in mineral oil and therefore had a final concentration of approximately 6.25%). Odors were presented for 2 s with an interstimulus interval of 40 s with 3–5 times repetitions. The order of odor delivery was not randomized. A 'no odor' trial with the same parameters but in which no odor valve opened was included with each set of repetitions. Odors were delivered through a mask with balanced input and output air flow that also allowed us to record respiration (*Grimaud and Murthy, 2018*). The positioning of the mask was adjusted daily to ensure optimal signal to noise. A photoionization detector (miniPID, Aurora Scientific) was used to confirm that odor concentrations were consistent between trials with these parameters. Odors were replaced before each set of experiments.

## In vivo imaging analysis (*Figures 2*, *3* and *4*)

Data were analyzed offline using custom-written scripts in MATLAB (Mathworks). Experimenters were blind to fluorescence changes during data analysis but not to experimental group.

## Regions of interest (ROIs)

Dendritic ROIs from abGCs were chosen based on average intensity projections in the dTomato channel as previously described (*Wallace et al., 2017*). Fields of view were non-overlapping and separated by at least 100 μm to minimize the chance of the same dendrites appearing in multiple fields of view. Z stacks of each imaging region were taken with a 2 μm z step from the surface of the dura down to the convergence of GC dendrites into a single apical dendrite. The density of labeling

precluded tracing of all dendrites back to their parent cell, but we used these z stacks to ensure that multiple ROIs were not chosen from the same dendrite. Therefore, these data should be considered a sample from a population of dendrites rather than cells since some dendrites may have originated from the same cell. We previously estimated with our labeling and imaging strategy that about 1.4 times more dendrites than cells are represented in the dataset (and this is almost certainly an overestimate since this was based on the first week of imaging, where labeling is significantly sparser than at later timepoints). For ablation after development (*Figure 4*), the same fields of view were imaged before and after PLX treatment. For *Figure 4—figure supplement 2*, we chose matching ROIs for the two sessions (any ROIs without similar morphology between the two sessions were excluded as described previously *Wallace et al., 2017*) to quantify how much our imaging conditions changed between sessions without PLX treatment. However, we opted to choose ROIs independently for *Figure 4* since matching ROIs always results in the exclusion of many ROIs, reducing sample size.

## Motion correction

To correct for fast lateral motion and image drift, all image frames for a given field of view were aligned to the average of the first trial using cross-correlation based on rigid body translation (ImageJ plugin Moco)(*Dubbs et al., 2016*). Frames with out-of-frame motion were removed based on the cosine similarity between each frame and the average intensity projection of the first trial (or the user-determined trial with the least motion for awake imaging). Image frames with cosine similarity that differed by more than 25% (30% for awake imaging) from the mean value for the user-determined best trial for that field of view or more than 20% (15% for awake imaging) from the mean of the baseline period for that trial were discarded. In some trials, immersion water dried up or laser power fluctuated, so trials were removed if their average brightness was less than half of the brightness of the average of the first three trials or if difference in brightness between the baseline and odor periods was greater than three times the standard deviation of brightness in the first trial. The entire trial was removed if, after these corrections, it contained less than 75% of the original frames during either the baseline or odor analysis period.

## Fluorescence changes

The average intensity in the GCaMP channel was calculated for each ROI, for each frame and for each odor. A response value for each cell-odor pair was calculated as the average $\Delta F/F_\sigma$ value over the odor analysis period (5 s following odor onset) where $F_\sigma$ represents the standard deviation of fluorescence during the baseline period. We used $F_\sigma$ because we found that in many cases the baseline GCaMP6s fluorescence was so low in abGC dendrites that we could not reliably subtract the background as described previously (*Wallace et al., 2017*). Bleaching was corrected by fitting a single exponential to the florescence during the baseline period and taking the value at the end of the baseline period as the baseline mean, only for ROIs where the fluorescence during the last 2.5 s of the baseline period was greater than 1.1 times the fluorescence during the first 2.5 s. If, after correction for bleaching, baseline $F_\sigma$ was greater than 30% of the mean baseline fluorescence, that ROI was considered too noisy and was removed from the analysis.

## Event detection (*Figure 3—figure supplement 1*)

Events were detected separately in each trace using the calculated ROC threshold (see next section) and any frames that were included in an event in any repeat were averaged across all repeats to obtain a mean event trace and the mean value in a 1 s period around the peak in this mean trace was then calculated as the average response to that odor. For latency, the mean latency across repeats was calculated for all repeats that had detected events.

## Thresholds

For all figures where a threshold was applied to the data, thresholds were calculated based on the distribution of 'no odor' trials. An area under the receiver operating curve analysis was performed and the lowest threshold yielding a 10% false positive rate was chosen. Thresholds were calculated for each figure by combining responses from both control and PLX-treated groups and performing ROC analysis on the combined data. For event detection, we used ROC analysis to find the optimal

combined threshold for the number of frames and standard deviation above baseline and chose the threshold that gave closest to a 10% false positive rate.

## Lifetime sparseness

After applying a threshold to the data, we used the following equation to calculate lifetime sparseness: (*Willmore and Tolhurst, 2001*)

$$LS = \frac{\left(\sum_{j=1}^{m} \frac{r_j}{m}\right)^2}{\left(\sum_{j=1}^{m} \frac{r_j^2}{m}\right)}$$

where m = number of odors, $r_j$ = response of the neuron to odor j.

If all stimuli activate a cell nearly uniformly, LS will be close to 1, and if only a small fraction of the stimuli activate a cell significantly, LS will be close to 0. For any cells with all responses below threshold, we set LS = 0, interpreting this as the sparsest possible representation.

## Temporal dynamics

Principal component analysis of the time course of responses was performed in Matlab using centered data and singular value decomposition as described previously (*Wienisch and Murthy, 2016*). To compare time courses for the control and PLX-treated groups, principal components were calculated on each dataset (all traces from all cell-odor pairs) separately and the angle between the two spaces spanned by the coefficient vectors for the first three principal components from the beginning to the end of the odor analysis period was calculated. Then a permutation test was performed in which the group to which each trace belonged was shuffled 1000 times, and the angles between new coefficient vectors were calculated based on random division into two groups of the same size as the original datasets. The actual angle was then compared to this distribution to obtain a p value.

## Respiration measurements

Peaks were extracted from respiration traces using the findPeaks function in Matlab with a minimum peak distance of 10 Hz.

## Raincloud plots

Raincloud plots of the type in *Figure 2G* were created using the Matlab version of the RainCloud-Plots package (*Allen et al., 2018*).

## Microglial depletion with CSF1R inhibitor PLX5622

CSF1R inhibitor PLX5622 was generously provided by Plexxikon (Berkeley, CA) and mixed into standard rodent diet at 1200 mg per kilogram of chow (Research Diets: AIN-76A diet). Control diet was formulated identically, but without the inhibitor.

## Flow cytometry (*Figure 2—figure supplement 1*)

Microglial depletion was confirmed via flow cytometry. A single-cell suspension enriched for microglia was generated as previously described (Hammond et al., Immunity 2019). Briefly, mice were deeply anesthetized with a mixture of ketamine (100 mg/kg) and xylazine (10 mg/kg) and transcardially perfused with 20 mL of cold Hank's balanced salt solution (HBSS, GIBCO, 14175–079). Bulbs and brains were minced using a razor blade (Electron Microscopy Science, 71960) and homogenized using a dounce tissue grinder (Wheaton, 357542). Microglia were enriched via centrifugation in 40% Percoll (Sigma-Aldrich, 17-0891-01) at 500 g for 1 hr at 4°C. Samples were incubated for 20 min with Blue Dead Cell Stain (Thermo Fisher, L34961) and Fc blocking antibody (Rat Anti-Mouse CD16/CD32, BD Bioscience, 553141) in HBSS + 2 mM EDTA. Cells were additionally stained for 20 min with antibodies against CD45 (Biolegend, 103116) and CD11b (Biolegend, 101217) in buffer (HBSS + 2 mM EDTA + 0.5% BSA).

Counting Beads (CountBright, Thermo Fisher Scientific, C36950) were added and samples analyzed using a FACS Aria II 'SORP'. All events were collected until a total of 8000 counting beads had been acquired for each sample. The data were analyzed in FlowJo 10.2.

### BrdU injections (*Figure 2—figure supplement 4*)

Mice received two intraperitoneal injections of BrdU (Sigma, 100 mg/kg in 0.9% saline) 12 hr apart.

### Fixed tissue preparation

Mice were deeply anesthetized with a ketamine/xylazine mixture and perfused transcardially with 20 mL of PBS (pH 7.4) first, followed by 30–50 mL of 4% paraformaldehyde (diluted from 16% stock, Electron Microscopy Sciences) in 0.1 M phosphate buffered saline (pH 7.4). Brains were removed from the skull and placed in 5 ml 4% paraformaldehyde for 2 hr. They were then rinsed with PBS and one hemisphere for each mouse was sliced coronally at 100 μm with a vibratome (Leica) for imaging of dTomato-labeled abGC spines (Figure 5), while the other hemisphere was sliced sagitally at 35–40 μm for immunostaining.

### Immunohistochemistry

For Iba-1 staining, 3–4 slices per mouse spanning the olfactory bulb were permeabilized and blocked with a solution containing 0.1% Triton X-100 (Fisher), and 5% goat serum in PBS for 1 hr at room temperature or blocked with Starting Block (ThermoFisher) with 0.3% TritonX-100 for 1 hr at room temperature and then incubated overnight at 4°C with the primary antibodies rabbit anti-Iba-1 (Wako: 019–19741, RRID:AB_839504) at 1:500 or rabbit anti-GFAP (Dako: Z0334, RRID: AB_10013382) at 1:1000 and then secondary antibodies (Alexa goat-647 anti-Rabbit) for 2 hr at room temperature. For BrdU/NeuN staining, one of every eight slices per mouse was chosen. Slices were washed in PBS with 0.1% Triton X three times for five minutes each before being incubated for in 2N HCl for 10 min at room temperature and then 20 min at 37°C. They were then placed in 0.1M Boric Acid buffer for 15 min and washed again three times with PBS. All slices were then incubated in starting block (ThermoFisher) with 0.3% Triton X for one hour before being staining in in PBS with 0.3% Triton X with rat anti-BrdU (Abcam: 6326 at 1:200), and mouse anti-NeuN (Millipore: MAB377 at 1:200) primary antibodies for 36–48 hr at 4°C and then secondary antibodies (Alexa Fluor 488 and 594 at 1:200) for 2 hr at room temperature. Slices were treated with 0.2% w/v Sudan Black in 70% EtOH for 5 min before mounting.

### Confocal imaging and quantification

Slices were mounted with DAPI mounting media (Vectashield DAPI) and imaged with a confocal microscope (LSM 880, Zeiss). Reported cell densities were calculated based on distances in fixed tissue, uncorrected for volume changes due to fixation and mounting. All imaging and quantification were performed blind to the experimental group of the animal (PLX-treated or control).

### Iba1 and GFAP quantification (*Figure 2—figure supplements 2* and *3*)

For the 1- and 4 week timepoints, one z-stack per animal was imaged at 10X with pixel size 0.42 × 0.42×1 μm spanning the thickness of the slice. For the 9 week timepoint, two z-stacks per animal were imaged at 20X and the counts from both were averaged. Stacks were imaged with pixel size 0.59 × 0.59×1 μm spanning 10 μm and converted to maximum intensity projections. The polygon tool was used to outline the granule cell layer in each image and the area was measured. Iba-1 or GFAP positive cells were counted in this area manually using the Cell Counter plugin on maximum intensity projection images in ImageJ. Cells were counted only if the cell body was fully included in the image stack.

### BrdU quantification (*Figure 2—figure supplement 4*)

For BrdU/NeuN, two z-stacks per OB (one centered dorsally and one centered ventrally) were taken at 20X with pixel size 0.52 × 0.52×0.89 μm spanning 9.8 μm. BrdU counts were performed using the automatic spots function in Imaris (Bitplane) with the same quality settings for spot detection for all images (quality threshold 2370, number of voxels threshold 524). Cells were counted as positive if they were located in the granule cell layer and were also positive for NeuN. The area of the granule cell layer in each image was measured in ImageJ, and the density of BrdU/NeuN positive cells was calculated for each image and averaged for all images for each mouse.

## Spine quantification (*Figure 5*)

All images were 20–40 µm z-stacks with 0.42 z-step taken with a 63x oil immersion objective, and four fields of view were imaged per mouse. Care was taken to ensure minimal saturation (less than 5% of pixels). Secondary and tertiary apical dendrites that were judged to be sufficiently bright, well-separated from adjacent or overlapping dendrites, at least 40 µm long, and extending at an angle from the imaging plane less than 45 degrees were chosen for tracing in Imaris. Each dendrite and all its dendritic protrusions less than 10 µm in length were manually traced using the Filaments function. Data from each dendrite were exported in a text file and imported into Matlab for plotting. For spine density, the Imaris property 'Filament No. Spine Terminal Pts' was used (meaning that branched spines were counted by the number of spine heads rather than attachment points). For spine head volume, the Imaris property 'Spine Part Volume Head' was used. For the classification of filopodial and mushroom spines (*Figure 5—figure supplement 1*, we used the Imaris properties 'HeadMaxDiameter' and 'NeckMeanDiameter' and classified spines as filopodial if they had maximum head diameter less than 1.5 times the neck mean diameter. This threshold was chosen because it led to a similar ratio of mushroom to filopodial spines in controls as previously observed (*Breton-Provencher et al., 2014*).

## Electrophysiology (*Figures 6* and *7*)

Two experiments with two littermate pairs each (control and PLX-treated) were performed for electrophysiology. Mice were deeply anesthetized with a mixture of ketamine (100 mg/kg) and xylazine (10 mg/kg) and perfused with ice-cold modified ACSF solution (in mM: 120 choline chloride, 25 glucose, 25 NaHCO3, 2.5 KCl, 0.5 CaCl2, 7 MgSO4, 11.6 ascorbic acid, 3.1 pyruvic acid, 1.25 NaH2PO4). Brains were removed and placed in the same ice-cold modified ACSF. Coronal slices (300 µm thick) of olfactory bulbs were cut using a vibratome (VT1000S; Leica, Germany). Slices were incubated in oxygenated holding solution (in mM: 119 NaCl, 26.2 NaHCO3, 1 NaH2PO4*H2O, 2.5 KCl, 22 glucose, 1.3 CaCl2, 2.5 MgSO4) at 33˚C for at least 30 min before being transferred to oxygenated ACSF (in mM: 119 NaCl, 26.2 NaHCO3, 1 NaH2PO4*H2O, 2.5 KCl, 22 glucose, 2.5 CaCl2, 1.3 MgSO4). Extra slices were maintained in holding solution at room temperature. Whole-cell recordings (Bessel filtered at 2.2 kHz and acquired at 10 kHz) were performed using patch pipettes filled with internal solution (in mM: 130 cesium gluconate, 5 NaCl, 10 HEPES, 12 phosphocreatine di (tris) salt, 3 Mg*ATP, 0.2 Na*GTP, 1 EGTA, 2.0 Na2-ATP, 0.5 Na3-GTP, pH 7.3) using a Multiclamp 700B amplifier (Molecular Devices, Palo Alto, CA) at 36˚C. Cells were visualized with dTomato and DIC with custom-built optics on a BX51WI microscope (Olympus Optical, Tokyo, Japan) and recorded with pClamp 10.3 (Molecular Devices). Cell identity was confirmed by the presence of fluorescence material in the patch pipet after membrane rupture and/or cell fill with Alexa Fluor 488, and only cells that had a proximal dendrite that extended from the soma in the direction of the EPL while remaining beneath the surface of the slice (i.e. cells that had dendrites that did not appear to have been cut during slicing) were targeted for patching. Patch pipets had 8–11 MΩ open tip resistance. Series resistance was not compensated. Cells were recorded in continuous 10 s sweeps for five minutes with a test pulse at the beginning of every sweep, which was used to calculate series resistance and holding current in Matlab for initial quality control. sEPSCs were recorded at −70 mV and sIPSCs were recorded at 0 mV. We waited at least one minute after breaking into the cell before beginning recording for sEPSCs and at least 30 s after switching the holding potential for sIPSCs; sEPSCs were always recorded first. Cells were recorded within 8 hr of slicing, and there did not appear to be any relationship between the time of recording and the frequency of synaptic events (data not shown). Experimenters were not blind to experimental group during recording.

### Electrophysiology analysis

Experimenters were blind to experimental group during analysis. Cells with an initial series resistance of <50 MΩ were used for analysis. Sweeps that deviated from the average series resistance across the first three sweeps by more than 25% or had a holding current of more than 100 pA at −70 mV were excluded. The cell was excluded entirely if less than half of the recording sweeps remained after quality control. For each cell, separate test pulses (−10 mV, 20 ms) with 50 repetitions at 20 kHz were recorded before and after each set of recordings, and these files were used to calculate series resistance (based on the maximum current recorded at the beginning of the pulse), membrane

resistance (based on the steady state current during the last 20% of the pulse), and cell capacitance (based on the time constant of an exponential fit between 20% and 80% of the current decay). Reported values in *Figure 6—figure supplement 1* and *Figure 7—figure supplement 1* are the means of these parameters from before and after the 5 min of sEPSC recordings. For sEPSCs and sIPSCs, all sweeps were concatenated for each cell (excluding 0.5 s around the test pulse), filtered with a 60 Hz band-stop filter with five harmonics in Matlab, and exported into Mini Analysis v. 6.0.7 (Synaptosoft). sEPSCs were detected with the following parameters: threshold 5, period to search a local minimum 10000, time before a peak to baseline 15000, period to search a decay time 20000, fraction of peak to find a decay time 0.37, period to average a baseline 1000, area threshold 20, number of points to average for peak 1, direction of peak 'negative.' After the initial detection step, 'Scan and detect double peaks' was selected. sIPSCs were detected with the following parameters: threshold 8, period to search a local minimum 10000, time before a peak to baseline 6000, period to search a decay time 20000, fraction of peak to find a decay time 0.37, period to average a baseline 1000, area threshold 80, number of points to average for peak 1, direction of peak 'positive.' In both cases, the detection was manually inspected, and the timepoints spanning any sections of the trace that exhibited increased noise (typically due to fluctuations in seal quality that caused many obviously false positive events) were noted. Event data was exported to a text file and imported into Matlab and noisy sections were excluded before further analysis. Cells that had many noisy sections were excluded. This included 3 control cells and 2 PLX cells for EPSCs and 2 control cells and 4 PLX cells for IPSCs (*Figure 6*) as well as 3 control and 3 PLX cells for EPSCs and eight control cells and 9 PLX cells for IPSCs (*Figure 7*). Events that were less than 3 ms apart were considered doubly detected and excluded.

## Statistical analysis

The number of mice to be used in each experiment was determined in advance, and all data that could be obtained from these mice under the experimental conditions and quality standards described in each section above were included in the analyses. To determine the number of mice for two-photon imaging, we took into account our previous success rate with combined cranial windows/virus injections (about 66%) and aimed to image a similar number of mice as in our previous work (4–7 mice for different experiments, *Wallace et al., 2017*). For electrophysiology, we aimed to obtain data from at least 20 cells per group from 3 to 4 mice, consistent with other electrophysiological investigations in the field that were able to reveal effects of various manipulations (*Pallotto et al., 2012*; *Quast et al., 2017*). Littermates were randomly allocated into control or PLX-treated groups for all experiments. Raw data are shown in every figure, in the form of a scatterplot overlaid on a bar graph when N < 40 or as a cumulative distribution otherwise. Non-parametric statistical tests were used for all comparisons, so no assumption of normality of data was made. Details for each statistical test can be found in the corresponding figure legend, and the rationale behind each is described here. The two-tailed Wilcoxon rank sum test was used to compare the medians of all unpaired non-normal distributions and correspondingly, the medians are plotted with bars in the figures. A one-tailed permutation test was used to test whether means in the real data were significantly higher than those in offset imaging data in *Figure 1*. A two-tailed permutation test was used to test for differences in principal components for the analysis of response time courses in *Figure 3—figure supplement 1* (details in section above 'Two-photon imaging, Temporal dynamics'). The hierarchical bootstrap (*Carpenter et al., 2003*; *Saravanan et al., 2019*) was used to compare groups in the case of hierarchical datasets (details below in section 'Hierarchical bootstrap'). The statistical test and p value for each test is found in the legend corresponding to each figure letter. All n values for each statistical test can be found at the end of each figure legend. In the text, we report mean ± SEM for datasets where means were compared or median (interquartile range) for datasets where medians were compared. For comparisons with hierarchical bootstrap, we report sampling distribution mean ±sampling distribution standard deviation.

## Hierarchical bootstrap

Sampling was performed 1000 times for each dataset as described (*Saravanan et al., 2019*). Specifically, a sample was taken with replacement across the units of the upper level. For example, in the two-photon imaging data, a sample was taken of the individual mice that were imaged (the sample

size was equal to the number of mice). Next, a sample was taken with replacement across the lower level, in this case the dendrites that were imaged. We chose to sample an equal number of dendrites from each mouse, regardless of the original number of dendrites imaged, to ensure that each animal would be represented equally (since the number of dendrites originally imaged for each animal was related to experimental considerations such as the density of viral labeling in each animal). For this level, we chose a sample size (100 dendrites) slightly larger than the largest sample per unit in the original data to ensure a representative sample would be obtained from each mouse. We then report the mean and the standard error of the bootstrapped parameter (either mean, median, or proportion, which is stated in each case) across the 1000 samples. We calculate a p value directly, which represents the probability that the parameter for the control group is greater than that for the PLX-treated group (for cases where the parameter for the PLX-treated group is significantly greater, such as the proportion of dendrites that were unresponsive in the imaging data, we report $1 - p$ to avoid confusion with standard p values). The same procedure was performed to analyze datasets for spine morphology (upper level = dendrites, lower level = spines, sample size = 100 spines) and electrophysiology (upper level = cells, lower level = EPSCs or IPSCs, sample size = 5000 EPSCs or IPSCs).

## Acknowledgements

We thank Martin Wienisch, Joseph Zak, Vikrant Kapoor, and Julien Grimaud from the Murthy lab for assistance with equipment setup and training in relevant techniques. Thanks to Samuel Marsh and the members of the Murthy and Stevens labs for productive discussions. Thanks to Arnaud Frouin and Alanna Carey from the Stevens lab for assistance with cloning and production of lentiviral plasmids. We thank Chen Wang and the Viral Core at Boston Children's Hospital for production of lentivirus.

## Additional information

### Funding

| Funder | Grant reference number | Author |
|---|---|---|
| National Institute on Deafness and Other Communication Disorders | R01DC013329 | Venkatesh N Murthy |
| National Science Foundation | DGE1144152 | Jenelle Wallace |
| National Institute on Deafness and Other Communication Disorders | F31 DC016482 | Jenelle Wallace |
| National Institutes of Health | R01NS092578 | Beth Stevens |
| Broad Institute | Merkin Award | Beth Stevens |
| Harvard Center for Biological Imaging | Simmons Family Award | Jenelle Wallace |

The funders had no role in study design, data collection and interpretation, or the decision to submit the work for publication.

### Author contributions

Jenelle Wallace, Conceptualization, Formal analysis, Investigation, Methodology, Performed two-photon imaging, electrophysiology, confocal imaging, and spine quantification experiments; Julia Lord, Investigation, Performed spine quantification experiments; Lasse Dissing-Olesen, Formal analysis, Investigation, Performed flow cytometry experiments and analyzed flow cytometry data; Beth Stevens, Venkatesh N Murthy, Conceptualization, Supervision, Funding acquisition, Project administration

## Author ORCIDs

Jenelle Wallace (ID) https://orcid.org/0000-0002-9565-2751
Beth Stevens (ID) https://orcid.org/0000-0003-4226-1201
Venkatesh N Murthy (ID) https://orcid.org/0000-0003-2443-4252

## Ethics

Animal experimentation: All procedures were performed under approved protocols in accordance with the Harvard University Institutional Animal Care and Use Committee (IACUC protocol #29-20) under Animal Welfare Assurance Number A3593-01.

## Decision letter and Author response

Decision letter https://doi.org/10.7554/eLife.50531.sa1
Author response https://doi.org/10.7554/eLife.50531.sa2

## Additional files

### Supplementary files

• Transparent reporting form

### Data availability

All data generated or analyzed during this study are included in the manuscript and supporting files. Source data files have been provided for Figures 1-7.

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
