## [Decision Letter]

Thank you for submitting your article "Microglia are necessary for normal functional development of adult-born neurons in the olfactory bulb" for consideration by *eLife*. Your article has been reviewed by three peer reviewers, and the evaluation has been overseen by Gary Westbrook as the Senior and Reviewing Editor. The following individuals involved in review of your submission have agreed to reveal their identity: Amanda Sierra (Reviewer #3).

The reviewers have discussed the reviews with one another and the Senior Editor has drafted this decision to help you prepare a revised submission.

Summary:

The reviewers were interested in the topic and acknowledged the sophisticated approach to the issue of microglial-synaptic interactions. In particular the set of analyses were thorough and reasonable given the results. However, we were somewhat disappointed by the overall lack of clear mechanism (either tested or hypothesized) about what the microglia are actually doing in the OB, and with respect to abGC maturation/integration. The key points discussed between reviewers are included here, but the full original reviews are at the end of this letter. We urge the authors to fully address the concerns raised as we expect that a revised manuscript would require an additional round of review given the extent of the comments.

Essential revisions:

1) The authors must acknowledge that depleting microglia may have unintended side effects (including inflammation) and that many microglial functions (beyond spine surveillance) will be altered.

2) Please better validate the method used to quantify spine surveillance.

3) The dendritic segments used for quantitation need to be specified and with a sufficient sample size to allow robust interpretation.

4) Please explain the relationship of dendritic Ca^2+^ correlates with spine Ca^2+^ for these experiments.

5) Perhaps most importantly, a mechanistic explanation of how microglial surveillance leads to increase spine response to odors is needed.

Reviewer #1:

The manuscript by Wallace et al. applied sophisticated in vivo 2-p imaging system to demonstrate a critical role of microglia in regulating development of adult-born neurons presumably by interacting with mushroom spines. Importantly, they showed that microglia ablation reduces odor responses of the adult-born neurons but had no effect on mature neurons, which correlates with reduced amplitude of EPSCs and reduced spine volume in adult-born neurons. In general, the experiments are well-designed and properly performed. The data generally support the conclusion. The Discussion section is particularly impressive, as the authors have extensively discussed potential caveats and future directions. Overall, this manuscript is appropriate for publication in *eLife* pending addressing the following points.

Major points:

1) For spine analysis, the author claimed they picked 1-5 apical dendrites, but it is unclear where those dendritic segments come from. For instance, the spine density maybe different in proximal versus distal dendritic regions. To be consistent, authors should pick the same number of dendritic segments at similar dendritic locations. Furthermore, the number of adult-born neurons they have for spine analysis is considered low. More cells are required for analysis.

2) Authors studied the interaction between microglia processes and dendritic spines of the adult-born neurons. However, they performed Ca^2+^ imaging in dendrites when multiple distinct odors were introduced. It is unclear how dendritic Ca^2+^ signal correlates with spine Ca^2+^ signals upon odor administration.

3) Authors consider 5-6 week old neurons as adult-born neurons, and 3 month old neurons as mature neurons. The assumption is that they represent different stages of maturation. However, it is unclear whether the spine phenotype is different between 5-6 week old and 3 month old neurons. Specifically, how different the quantity and quality of the mushroom spines in 5-6 week and 3 month old neurons? Some basic quantification on the spine phenotype from the two time points need to be provided to support the rationale of choosing these two time points.

Reviewer #2:

Wallace et al. performed series of two photon imaging studies of microglial interactions with developing and mature adult born granule cells (abGCs) in the olfactory bulb. Their findings indicate that microglial processes preferentially interact with abGC spines, and when pharmacologically ablated, leads to decreased odor responsivity and amplitude of excitatory input onto abGCs. Overall, this study presents straightforward evidence that microglia in the olfactory bulb influence responsivity of adult-born granule cells to odors. Specifically, a strength of this study is their use of GCaMP imaging from abGC dendrites in vivo, which shows that the amplitudes of abGC responses to odors are selectively reduced when microglia are ablated during abGC development. The authors present this as a mechanism by which microglial ablation may lead an increase in principle cell odor responses. While the imaging of odor responses in abGC dendrites after microglial ablation is novel and compelling, the supporting evidence regarding microglial interactions with abGC spines and the functional significance (electrophysiology) raise some questions and concerns.

– The impact of this study would be greatly enhanced by some mechanistic explanation for how microglial contact with mushroom spines leads to an increased response to odors. The authors briefly speculate about pruning and stabilization, but it seems like it is well-within the capabilities of the authors to directly test the pruning hypotheses. Or is this developmental stalling? On the other hand, the impact would also be enhanced by demonstrating a behavioral phenotype after microglial ablation. For example, the authors speculate at the end of the Discussion that "…microglial regulation of (abGC) development may contribute to ongoing plasticity in the olfactory system." This could certainly be tested similarly as in the papers cited as supporting evidence.

– Although many appropriate analyses were performed, in general the results show relatively small effects, and correlations at times come across a bit overinterpreted. For example, the effect observed regarding microglial interactions with mushroom spines seems trivially small considering the weight that is put on it throughout the paper. Can the differences in average interaction time and number of interactions alone explain the larger effect on microglial ablation on abGC odor responses? The same criticism applies to the interpretation of the spine head volume after ablation and altered mEPSC amplitudes. Is it not also likely that effects of microglial ablation on other cell types in the OB (or throughout the brain) lead to changes in abGC recruitment?

– Did the authors perform statistics on all the CD plots in Figure 1? Are none significantly different? This applies to all CD plots where significance is not noted throughout the paper.

– Figure 7—figure supplement 1B directly contradicts the authors' claim that microglia ablation after abGC maturation does not affect their synaptic input. In fact, this is one of the more compelling differences seen in the electrophysiological recording data. It seems to also directly contradict Figure 7C, where the only difference is that one shows median mEPSC frequency by cell (Figure 7C) and one shows mean mEPSC frequency (Figure 7—figure supplement 1B). It would be helpful to see a more in-depth characterization of mature abGC and resident GC inputs after microglial ablation, especially given the open-ended nature of the discussion eluding to the possibility that the observed trends may indeed point to alterations in mature cells too.

– It appears that the heads of filopodial spines would be harder to define, and ROIs would be more likely to contain necks. Given the small effect seen in 1e, is it possible that any effect on non-mushroom spines is masked by the inclusion of dendrite and spine shafts?

– If similar colocalization analysis was performed on the entire visible abGC (including spines and dendrites together) is there a preferential interaction of microglia with whole abGC dendrites (compared to offset data)?

– The in vivo imaging data compares responses in populations of dendrites imaged from a small number of animals. The small number of animals is understandable given the nature of the experimentation, but nested statistics should be used considering that many data points are collected from the same animals (and even cells) within the same fields of view.

– The authors state that abGCs initially show broader tuning to odors that become refined over time and during development. It is a bit difficult to completely reconcile and/or understand this statement with respect to the accompanying data given the "sparser" responses observed with microglia ablation. Do the authors suggest the affected abGCs are more mature, stalled in development, selectively pruned, or just altered? These observations don't seem to fit with any "tuning" argument. In fact, other studies have reported broadening of tuning in both abGCs and developing cortical interneurons with maturation.

– To point above, and given the lack of mechanistic details presented here, it would be appreciated for the authors to at least provide some deeper insight and/or speculation within the discussion of what is actually happening with microglia ablation. Is this developmental stalling, failure for synaptic maintenance, selective survival, unchecked (and selective) pruning of spine subsets? As is, the data do not fully support a clear interpretation.

Reviewer #3:

The manuscript "Microglia are necessary for the normal functional development of adult-born neurons in the olfactory bulb" by Wallace et al., addresses a hot topic in neuroscience (the role of microglia in synaptic connectivity) using as a model adult neurogenesis in the OB. The topic is novel, the experimental design is sound, and the data supports the conclusions. There are a few issues that should be addressed prior to publication, mostly to reach non-specialized audiences.

Major issues:

1) My major criticism is on the method used to address the role of microglia, i.e., microglial ablation.

In the first place, as the authors succinctly mention in the Discussion, ablation has been associated to the development of inflammatory responses (in some paradigms, it even leads to a "cytokine storm"). The authors should clearly define the inflammatory conditions in their model of ablation in the OB (which, to the best of my knowledge, has not been defined yet).

In addition, dead microglia do not just "vanish" – surely, clearing the microglia debris may indirectly affect their results and this should at least be acknowledged in the methodological consideration section.

Finally, the authors seem to imply that ablating microglia solely affects their surveillance of mushroom spines and that thus this must be the only mechanism by which microglia acts on developing newborn neurons. The authors should discuss alternative mechanisms for the observed effect on the functional responses of the newborn neurons.

2) The method used to analyze the interaction between microglia and spines is rather obscure and does not seem to have been validated before, as no paper is cited either in the Results or the Materials and methods section. If it is not based on previously published methods, the authors must validate it in the context of this paper. To validate their method, the authors should test it with well known co-localizating objects and then test it with the microglia-spine co-localization.

My major concern is that the interaction of microglia with the spines is compared to an Offset image that is calculated by horizontally translating the image stack by half of the total pixels (subsection “Analysis of microglia-spine interactions”). However, using a single Offset (instead of an iteration of Offsets) may lead to strong biases when comparing with the actual data. An example of this potential issue is that the Offset for mushroom spines is 34% whereas is 24.6% for filopodial spines (subsection “Microglia preferentially interact with mushroom spines on abGCs” second paragraph). This difference could be resulting from different densities and relative sizes of the two types of spines; or to the fact that the Offset is not representative of all the possible Offsets.

A more proper control would be in fact to iterate small translations (pixel by pixel) or even rotations (degree be degree) and obtain a probability distribution of Offsets, which then could be compared with the actual data.

Furthermore, in data shown in Figure 1D, the statistical effect seems to be related to an outlier in the real data, not to an actual difference of distribution.

3) The authors only briefly mention a similar paper published in 2017 in *eLife* on microglial ablation and OB neurogenesis (Reshef et al., The role of microglia and their CX3CR1 signaling in adult neurogenesis in the olfactory bulb). To enhance the perception of novelty the authors should put more effort into clearly discriminating what has been done vs what is learned in this paper (Introduction, paragraph four).

4) The text is rather arid at times, particularly when describing results using non-conventional techniques, as the authors seem to assume all readers will be fully experts in their methodology.

For instance, in the sentence "To quantify whether microglia preferentially interact with dendritic spines (defined as colocalization of a microglial process with a spine head, meaning the two are within the diffraction limit of our microscope) on abGCs compared to encountering them by chance during the course of continuous motility" (subsection “Microglia preferentially interact with mushroom spines on abGCs”) the reference to the diffraction limit of the microscope is really puzzling.

Similarly, "lifetime sparseness" is not explained (subsection “Odor responses are reduced in abGCs that mature in the absence of microglia” paragraph two).

Principal component analysis is not explained at all (subsection “Odor responses are reduced in abGCs that mature in the absence of microglia” paragraph three) – which variables were used? (the responses to all 15 odors?).

The authors also state that they applied an "event detection analysis" (subsection “Odor responses are reduced in abGCs that mature in the absence of microglia” paragraph three), which is not explained whatsoever.

5) Some figures are difficult to read.

Figure 1: the parameter measured in each of D-L figures is at the bottom of each pair, and is in small size. So when reading figure 1D one has, in fact, to go to the bottom of Figure 1I to figure out what parameter is actually shown.

Figure 2: odors should be either labeled or numbered (and referenced with names in the figure legend). There is a reference to Table 1 but this reviewer has not been able to find it.

Figure 3—figure supplement 1 shows the same odor and odor mean traces as Figure 3A. Both figures should be combined. In addition, the traces seem to lack asterisks denoting statistical significance.

Figure 3—figure supplement 2 is extremely puzzling. What do X and Y-axes represent in A and B? the eigenvalues of the PCs? The authors should show the 3D plot PC 1 vs PC 2 vs PC 3 showing whether anesthetized and awake mice group separately.

6) In addition to the number of odors generating responses in anesthetized vs awake mice (Figure 2—figure supplement 2 vs Figure 3A) there seems to be also a quality effect: not the same odors generate responses. The authors should comment on this issue.

7) The authors should check an efficient microglial ablation in the "after development" experiment (Figure 4). They should also speculate why developing neurons seem more susceptible than mature neurons in their regulation by microglia.

8) The authors conclude that "In the absence of microglia, these spines have reduced volume, and this corresponds to weaker excitatory synapses." The second part of the sentence (whether synapses are weaker in the absence of microglia) has not been directly tested in the paper. The authors should either state that they are speculating, or test it directly.

---

## [Author Response]

Essential revisions:1) The authors must acknowledge that depleting microglia may have unintended side effects (including inflammation) and that many microglial functions (beyond spine surveillance) will be altered.

To address this important point, we have added Figure 2—figure supplement 3 which shows that the number of GFAP+ cells in the OB does not change after PLX ablation, supporting previous work (Elmore et al., 2014 and Reshef et al., 2017) that showed that PLX ablation, in contrast to diphtheria toxin-mediated ablation (Bruttger et al., 2015), does not cause widespread inflammation characterized by large increases in the number of GFAP+ astrocytes and a cytokine storm. However, we acknowledge that this does not preclude other undetectable effects on the inflammatory environment, and we have expanded our discussion on this point (in the Discussion section, Methodological Considerations). We have also added to our discussion of possible alterations in microglial function beyond spine surveillance (Discussion, Microglia-neuron interactions in the adult brain) and modified the title of the paper to more accurately represent our results.

2) Please better validate the method used to quantify spine surveillance.

We appreciate the feedback on this point, and have significantly improved our method for analyzing the two photon imaging data on microglia-abGC interactions as the reviewer suggested. Instead of using a single image-offset control on the imaging data, we now perform a series of offsets and construct probability distributions for the measured parameters. The reanalyzed data is shown in Figure 1 and largely supports our original analysis, confirming the increased number of interactions with mushroom spines.

3) The dendritic segments used for quantitation need to be specified and with a sufficient sample size to allow robust interpretation.

We have specified that spines were quantified on secondary and tertiary abGC apical dendrites in the EPL (Results, Synapse development in abGCs that mature in the absence of microglia and Materials and methods, Confocal imaging and quantification, Spine quantification and box in the diagram in Figure 5A). In addition, we increased our sample size for this experiment by tracing dendrites from additional cells. The extended data set supports our original conclusions, with the differences in spine density now significant. We have also added Figure 5—figure supplement 1 to explore the differences in spine density and morphology in control vs. PLX-treated abGCs in more detail.

4) Please explain the relationship of dendritic Ca^2+^ correlates with spine Ca^2+^ for these experiments.

We have clarified this relationship and our choice to measure dendritic Ca^2+^ in abGCs (Results, Odor responses are reduced in abGCs that mature in the absence of microglia and Discussion, AbGCs in the olfactory circuit). We explain further here: Single spine Ca^2+^ transients have not been systematically observed in vivo in granule cells, and the relationship between dendritic and spine calcium and the functional significance of possible compartmentalization has been investigated in slice preparations but not in vivo in these cells. Therefore, we chose to record dendritic calcium events which likely represent dendritic calcium spikes or global calcium events accompanying somatic action potentials that result from simultaneous activation of many feedforward and feedback synapses, which are the most well-characterized patterns of activity in vivo.

5) Perhaps most importantly, a mechanistic explanation of how microglial surveillance leads to increase spine response to odors is needed.

We interpret this point as a request to explain in the text how microglial surveillance may lead to altered functional response of abGCs, rather than a request to perform additional mechanism-seeking experiments (which are difficult and open-ended). We have expanded our discussion on this point (“AbGCs in the olfactory circuit” and “Microglia-neuron interactions in the adult brain”) and added Figure 8 to more completely summarize our results and explain our model.

Reviewer #1:[…]1) For spine analysis, the author claimed they picked 1-5 apical dendrites, but it is unclear where those dendritic segments come from. For instance, the spine density maybe different in proximal versus distal dendritic regions. To be consistent, authors should pick the same number of dendritic segments at similar dendritic locations. Furthermore, the number of adult-born neurons they have for spine analysis is considered low. More cells are required for analysis.

We agree with the reviewer. We have now specified that spines were quantified on secondary and tertiary abGC apical dendrites in the EPL (Results, “Synapse development in abGCs that mature in the absence of microglia” and Materials and methods, “Confocal imaging and quantification”, “Spine quantification” and Figure 5A). In addition, we increased our sample size for this experiment by tracing dendrites from additional cells. The extended data set supports our original conclusions, with the differences in spine density now significant. We have also added Figure 5—figure supplement 1 to explore the differences in spine density and morphology in control vs. PLX-treated abGCs in more detail.

2) Authors studied the interaction between microglia processes and dendritic spines of the adult-born neurons. However, they performed Ca^2+^ imaging in dendrites when multiple distinct odors were introduced. It is unclear how dendritic Ca^2+^ signal correlates with spine Ca^2+^ signals upon odor administration.

We have clarified this relationship and our choice to measure dendritic Ca^2+^ in abGCs (Results, “Odor responses are reduced in abGCs that mature in the absence of microglia” and Discussion, “AbGCs in the olfactory circuit”). We explain further here: Single spine Ca^2+^ transients have not been systematically observed in vivo in granule cells, and the relationship between dendritic and spine calcium and the functional significance of possible compartmentalization has been investigated in slice preparations but not in vivo in these cells. Therefore, we chose to record dendritic calcium events which likely represent dendritic calcium spikes or global calcium events accompanying somatic action potentials that result from simultaneous activation of many feedforward and feedback synapses, which are the most well-characterized patterns of activity in vivo.

3) Authors consider 5-6 week old neurons as adult-born neurons, and 3 month old neurons as mature neurons. The assumption is that they represent different stages of maturation. However, it is unclear whether the spine phenotype is different between 5-6 week old and 3 month old neurons. Specifically, how different the quantity and quality of the mushroom spines in 5-6 week and 3 month old neurons? Some basic quantification on the spine phenotype from the two time points need to be provided to support the rationale of choosing these two time points.

We see the point of confusion raised by the reviewer and have improved the clarity in our text. We do not intend to perform a direct comparison between 5-6 weeks old vs. 3 month old abGCs. We chose 5-6 week old abGCs as a relatively “mature” timepoint (based on our previous in vivo imaging data in Wallace et al., 2017) at which we might expect to see the outcome of differences occurring through development with PLX ablation. We chose the 3 month-old stage as a timepoint at which we could be sure maturation would be completed to test the effect of PLX ablation after development. However, we expect that the exact timepoint would not matter because we previously showed that there was no difference in dendritic calcium responses between mature abGCs (5-6 weeks old) and “resident” GCs that were 2-3 months old at the time of imaging (Wallace et al., 2018, Supplementary Figure 5). We have modified the wording (Results, “Microglia ablation after development has no effect on odor responses”) to clarify this.

Reviewer #2:[…]– The impact of this study would be greatly enhanced by some mechanistic explanation for how microglial contact with mushroom spines leads to an increased response to odors. The authors briefly speculate about pruning and stabilization, but it seems like it is well-within the capabilities of the authors to directly test the pruning hypotheses. Or is this developmental stalling?

We agree with the reviewer that it is important to investigate possible mechanistic explanations. We did in fact test one possible mechanism relating to microglia-neuron interactions, complement C1q-mediated synaptic pruning (Author response image 1). We are happy to include these data in the paper if the reviewer and editor prefer. However, ruling out one pathway does not preclude the possibility of pruning occurring in this system. Unfortunately, it is more difficult to test pruning here than it might seem at first glance – unlike during early development, when entire brain regions experience waves of pruning during defined timepoints, a very small percentage of the total number of granule cells would be at this stage at any given time. Furthermore, most of our (Stevens) lab’s previous work points to developmental microglia-mediated pruning occurring on the pre- rather than post-synaptic side, and there is no straightforward way to selectively label and distinguish the inputs to adult-born granule cells (mitral cells and cortical feedback would be expected to synapse onto both adult-born and pre-existing granule cells). Therefore, we feel that addressing mechanisms of pruning would require a large number of experiments, well beyond the scope the present study.

**Author response image 1. respfig1:** Spine density and volume are unaffected in C1qKO mice. (**a**) Experimental timeline for spine quantification in wild type versus C1qKO mice. Lentivirus was injected into the RMS to label abGCs. Spine numbers and morphology were quantified 4 weeks later. (**b**) Sample image showing sections of apical dendrites from an abGC in a control mouse (top) and a C1qKO mouse (below). (**c**) Spine density averaged in apical dendrites of abGCs. (**d**) Cumulative distribution showing the volume of all spines analyzed. Inset, head volume averaged across all spines in each cell. Bars indicate medians across cells (circles). n = 1215 spines from 26 abGCs from 3 wild type mice and n = 1262 spines from 27 abGCs from 3 C1qKO mice.

On the other hand, the impact would also be enhanced by demonstrating a behavioral phenotype after microglial ablation. For example, the authors speculate at the end of the Discussion that "…microglial regulation of [abGC] development may contribute to ongoing plasticity in the olfactory system." This could certainly be tested similarly as in the papers cited as supporting evidence.

We did contemplate testing whether there is a behavioral effect of microglia ablation. However, we opted not to perform these experiments because we felt that the results would be difficult to interpret. Any effect that we might find could not be unambiguously attributed to deficits in abGCs in the olfactory bulb because microglia ablation in our experiments is brain-wide, and it is likely that many olfactory regions are also necessary for the types of olfactory tasks that have been shown to require abGCs. Currently we do not have a way to ablate or manipulate microglia in the olfactory bulb or any specific brain region during a defined time window in adult animals.

– Although many appropriate analyses were performed, in general the results show relatively small effects, and correlations at times come across a bit overinterpreted. For example, the effect observed regarding microglial interactions with mushroom spines seems trivially small considering the weight that is put on it throughout the paper. Can the differences in average interaction time and number of interactions alone explain the larger effect on microglial ablation on abGC odor responses? The same criticism applies to the interpretation of the spine head volume after ablation and altered mEPSC amplitudes. Is it not also likely that effects of microglial ablation on other cell types in the OB (or throughout the brain) lead to changes in abGC recruitment?

We agree that the effects on spine head volume and sEPSC amplitude are small, but we argue that changes in calcium responses could easily be larger because dendritic calcium transients are nonlinear outcomes of simultaneous activation of many synapses. We have discussed this further (Discussion, “AbGCs in the olfactory circuit”). We also agree that microglial interactions with spines are only one aspect of microglial function that may be altered when microglia are ablated and we discuss this further in our manuscript (Discussion, “AbGCs in the olfactory circuit” and Discussion, “Microglia-neuron interactions in the adult brain”).

– Did the authors perform statistics on all the CD plots in Figure 1? Are none significantly different? This applies to all CD plots where significance is not noted throughout the paper.

We modified the analysis for Figure 1 according to Reviewer 3’s comments (see response to essential point 2) so this figure no longer shows CD plots.

In general, in most cases we originally showed CD plots to show the cumulative distribution of an entire dataset (for example, all EPSCs from all cells), in the spirit of showing raw data rather than only summary statistics. However, we felt that it would be misleading to perform statistics on these plots (using a Kolmogorov-Smirnov test, example) because the data points are not independent (for example, there are hundreds of sEPSCs recorded from each cell). To deal with this issue, we now use the hierarchical bootstrap (Saravanan et al., 2019) to perform hypothesis testing on these types of hierarchical datasets throughout the paper, in cases where multiple data points are measured from each cell. As discussed in Saravanan et al., 2019, this method is more appropriate for hierarchical datasets (in terms of preserving statistical power and avoiding elevating the false positive rate) than either comparing cumulative distributions of all data points directly or comparing data summarized per unit (for example, taking the mean across each cell and comparing these between groups), which was our original approach. The modified analysis did not change the previous conclusions.

– Figure 7—figure supplement 1B directly contradicts the authors' claim that microglia ablation after abGC maturation does not affect their synaptic input. In fact, this is one of the more compelling differences seen in the electrophysiological recording data. It seems to also directly contradict Figure 7C, where the only difference is that one shows median mEPSC frequency by cell (Figure 7C) and one shows mean mEPSC frequency (Figure 7—figure supplement 1B). It would be helpful to see a more in-depth characterization of mature abGC and resident GC inputs after microglial ablation, especially given the open-ended nature of the discussion eluding to the possibility that the observed trends may indeed point to alterations in mature cells too.

We agree that this was unclear. The difference between the original Figure 7C and Figure 7—figure supplement 1B was that Figure 7C shows frequency calculated as inter-event frequency (so there will be one data point for each event) while Figure 7—figure supplement 1B showed average frequency across the entire recording period (total # of events divided by total time, i.e. one value for each cell). Related to the point above about the analysis of hierarchical datasets (Saravanan et al., 2019), we have removed Figure 7—figure supplement 1 and instead reanalyzed the data in Figure 7C with the hierarchical bootstrap. This new analysis shows that the same trend is present regardless of how frequency is calculated in this dataset (Figure 7C—the mean frequency is higher for PLX-treated cells), although it does not reach the threshold for statistical significance. We have reflected on this further in the Discussion, Timing of microglia ablation.

– It appears that the heads of filopodial spines would be harder to define, and ROIs would be more likely to contain necks. Given the small effect seen in 1e, is it possible that any effect on non-mushroom spines is masked by the inclusion of dendrite and spine shafts?

We drew ROIs on filopodial spines to be approximately circular with diameter equal to the width of the spine head, so it is unlikely that a substantial portion of the shaft was included, and ROIs did not include any portion of the parent dendrite. Without performing electron microscopy reconstructions, we can only do our best to estimate the location of the postsynaptic density for both types of spines.

– If similar colocalization analysis was performed on the entire visible abGC (including spines and dendrites together) is there a preferential interaction of microglia with whole abGC dendrites (compared to offset data)?

We did not perform this analysis because it is unclear to us how changes in microglial interaction with dendrites rather than synapses could be interpreted as this has not yet been explored in the field.

*– The* in vivo *imaging data compares responses in populations of dendrites imaged from a small number of animals. The small number of animals is understandable given the nature of the experimentation, but nested statistics should be used considering that many data points are collected from the same animals (and even cells) within the same fields of view.*

We agree that this is an important statistical problem that is often not adequately addressed in the field, and we sincerely thank the reviewer for pointing this out. This motivated us to apply the hierarchical bootstrap method to our data, as discussed above.

– The authors state that abGCs initially show broader tuning to odors that become refined over time and during development. It is a bit difficult to completely reconcile and/or understand this statement with respect to the accompanying data given the "sparser" responses observed with microglia ablation. Do the authors suggest the affected abGCs are more mature, stalled in development, selectively pruned, or just altered? These observations don't seem to fit with any "tuning" argument. In fact, other studies have reported broadening of tuning in both abGCs and developing cortical interneurons with maturation.

We agree with the reviewer that it is difficult to interpret the sparser odor responses in PLX-treated mice in light of our previous data showing the mature abGCs have sparser responses than young abGCs, primarily because the mechanisms that underlie normal developmental maturation of odor responses in these cells are still under active investigation. We have expanded our discussion of this point in the Discussion section, “AbGCs in the olfactory circuit”. In addition, we added Figure 8 to propose a model that integrates the results of the current study with our previous study (Wallace et al., 2017) and puts these findings in the context of other studies of abGC development. In short, we suggest that microglia ablation affects some aspects of abGC development (maturation and possibly selective reorganization of excitatory inputs) but not others (decrease in excitability and input resistance), leading to the phenotype we observe.

We note that it is difficult to compare maturation of abGCs with cortical interneurons, given their vastly different connectivity patterns in their respective circuits (for example PV interneurons are densely interconnected with excitatory neurons in both cortex and OB, but granule cells are much more sparsely connected than PV cells in the OB (Kato et al., 2013)).

– To point above, and given the lack of mechanistic details presented here, it would be appreciated for the authors to at least provide some deeper insight and/or speculation within the discussion of what is actually happening with microglia ablation. Is this developmental stalling, failure for synaptic maintenance, selective survival, unchecked (and selective) pruning of spine subsets? As is, the data do not fully support a clear interpretation.

We appreciate this comment, and it has motivated us to clarify our hypotheses about what is happening in this system by adding Figure 8, which summarizes our findings in the context of a potential model for how microglia might be acting on abGCs. As explained in our response to the previous point as well as in the Discussion section, “AbGCs in the olfactory circuit”, we suggest that microglia ablation affects some aspects of abGC development (maturation and possibly selective reorganization of excitatory inputs) but not others (decrease in excitability and input resistance), leading to the phenotype we observe.

Reviewer #3:[…]1) My major criticism is on the method used to address the role of microglia, i.e., microglial ablation.In the first place, as the authors succinctly mention in the Discussion, ablation has been associated to the development of inflammatory responses (in some paradigms, it even leads to a "cytokine storm"). The authors should clearly define the inflammatory conditions in their model of ablation in the OB (which, to the best of my knowledge, has not been defined yet).In addition, dead microglia do not just "vanish" – surely, clearing the microglia debris may indirectly affect their results and this should at least be acknowledged in the methodological consideration section.

To address this important point, we have added Figure 2—figure supplement 3 which shows that the number of GFAP+ cells in the OB does not change after PLX ablation, supporting previous work (Elmore et al., 2014 and Reshef et al., 2017) that showed that PLX ablation, in contrast to diphtheria toxin-mediated ablation (Bruttger et al., 2015), does not cause widespread inflammation characterized by large increases in the number of GFAP+ astrocytes and a cytokine storm. However, we acknowledge that this does not preclude other undetectable effects on the inflammatory environment, and we have expanded our discussion on this point (in the section Discussion, “Methodological considerations”).

Finally, the authors seem to imply that ablating microglia solely affects their surveillance of mushroom spines and that thus this must be the only mechanism by which microglia acts on developing newborn neurons. The authors should discuss alternative mechanisms for the observed effect on the functional responses of the newborn neurons.

We have added to our discussion of possible alterations in microglial function beyond spine surveillance (Discussion, “Microglia-neuron interactions in the adult brain”) and modified the title of the paper to more accurately represent our results.

2) The method used to analyze the interaction between microglia and spines is rather obscure and does not seem to have been validated before, as no paper is cited either in the Results or the Materials and methods section. If it is not based on previously published methods, the authors must validate it in the context of this paper. To validate their method, the authors should test it with well-known co-localizating objects and then test it with the microglia-spine co-localization.My major concern is that the interaction of microglia with the spines is compared to an Offset image that is calculated by horizontally translating the image stack by half of the total pixels (subsection “Analysis of microglia-spine interactions”). However, using a single Offset (instead of an iteration of Offsets) may lead to strong biases when comparing with the actual data. An example of this potential issue is that the Offset for mushroom spines is 34% whereas is 24.6% for filopodial spines (subsection “Microglia preferentially interact with mushroom spines on abGCs” second paragraph). This difference could be resulting from different densities and relative sizes of the two types of spines; or to the fact that the Offset is not representative of all the possible Offsets.A more proper control would be in fact to iterate small translations (pixel by pixel) or even rotations (degree be degree) and obtain a probability distribution of Offsets, which then could be compared with the actual data.Furthermore, in data shown in Figure 1D, the statistical effect seems to be related to an outlier in the real data, not to an actual difference of distribution.

We appreciate the reviewer’s detailed assessment of the method and agree with the concern that using a single Offset may not adequately represent a true shuffling of the data. Therefore, we have implemented the method the reviewer has suggested (iterating through many offsets).

Please see response to essential point 2 for further details.

3) The authors only briefly mention a similar paper published in 2017 in eLife on microglial ablation and OB neurogenesis (Reshef et al., The role of microglia and their CX3CR1 signaling in adult neurogenesis in the olfactory bulb). To enhance the perception of novelty the authors should put more effort into clearly discriminating what has been done vs what is learned in this paper (Introduction, paragraph four).

We have expanded our reference to this work both in the Introduction and in the Discussion to clarify that we have analyzed the functional effects of microglia ablation on abGCs, providing a mechanistic explanation for the reduction in mitral cell activity observed in Reshef et al., 2017.

4) The text is rather arid at times, particularly when describing results using non-conventional techniques, as the authors seem to assume all readers will be fully experts in their methodology.For instance, in the sentence "To quantify whether microglia preferentially interact with dendritic spines (defined as colocalization of a microglial process with a spine head, meaning the two are within the diffraction limit of our microscope) on abGCs compared to encountering them by chance during the course of continuous motility" (subsection “Microglia preferentially interact with mushroom spines on abGCs”) the reference to the diffraction limit of the microscope is really puzzling.Similarly, "lifetime sparseness" is not explained (subsection “Odor responses are reduced in abGCs that mature in the absence of microglia” paragraph two).Principal component analysis is not explained at all (subsection “Odor responses are reduced in abGCs that mature in the absence of microglia” paragraph three) – which variables were used? (the responses to all 15 odors?).The authors also state that they applied an "event detection analysis" (subsection “Odor responses are reduced in abGCs that mature in the absence of microglia” paragraph three), which is not explained whatsoever.

We appreciate the feedback on this point and have endeavoured to clarify the reasoning behind our use of these methods in the main text, with references to the Materials and methods section where appropriate.

5) Some figures are difficult to read.Figure 1: the parameter measured in each of D-L figures is at the bottom of each pair, and is in small size. So when reading Figure 1D one has, in fact, to go to the bottom of Figure 1I to figure out what parameter is actually shown.

We added labels to each individual panel to improve the readability of Figure 1.

Figure 2: odors should be either labeled or numbered (and referenced with names in the figure legend). There is a reference to Table 1 but this reviewer has not been able to find it.

Due to the length of some of the chemical names, we opted not to include these in the figure itself. Thank you for alerting us to the omission of the odor list, we have now included Table 1 in the Materials and methods under the section “Odor stimulation”.

Figure 3—figure supplement 1 shows the same odor and odor mean traces as Figure 3A. Both figures should be combined. In addition, the traces seem to lack asterisks denoting statistical significance.

Thank you, we have combined these figures. We did not include asterisks for these traces (similar to Figure 2E) because we did not test differences for individual odors (to avoid issues with multiple comparisons) and rather restricted significance testing to parts B, C, and D. In contrast, we show asterisks in Figure 2D denoting individual example responses above threshold.

Figure 3—figure supplement 2 is extremely puzzling. What do X and Y-axes represent in A and B? the eigenvalues of the PCs? The authors should show the 3D plot PC 1 vs PC 2 vs PC 3 showing whether anesthetized and awake mice group separately.

Thank you for pointing this out. We have clarified in the legend that the x-axis represents time and the y-axis represents ΔF/F_σ_. We have plotted it this way (as in Wienisch and Murthy, 2016 and Wallace et al., 2017) because a single 3D plot could not show the entire time course. As we have now clarified in the Materials and methods (“In vivoimaging analysis” and “Temporal dynamics”), we then used a permutation test to compare the time courses between control and PLX-treated mice for each condition.

6) In addition to the number of odors generating responses in anesthetized vs awake mice (Figure 2—figure supplement 2 vs Figure 3A) there seems to be also a quality effect: not the same odors generate responses. The authors should comment on this issue.

Previous studies (Kato et al., 2012; Wienisch and Murthy, 2016; Wallace et al., 2017) have described significant reorganization of responses in granule cells in awake vs. anesthetized mice, which was part of the rationale for imaging in both conditions. Specifically, there seems to be a weaker relationship between the number of glomeruli activated by each odor and the number of granule cells activated in awake mice compared to anesthetized mice (Wienisch and Murthy, 2016). We did not comment extensively on this in the paper because we thought it might distract from the difference between control and PLX-treated mice, but we could add more discussion if the reviewer prefers.

7) The authors should check an efficient microglial ablation in the "after development" experiment (Figure 4).

This is shown in Figure 2—figure supplement 2D.

They should also speculate why developing neurons seem more susceptible than mature neurons in their regulation by microglia.

We have discussed this in the Discussion section, “Timing of microglia ablation”.

8) The authors conclude that "In the absence of microglia, these spines have reduced volume, and this corresponds to weaker excitatory synapses." The second part of the sentence (whether synapses are weaker in the absence of microglia) has not been directly tested in the paper. The authors should either state that they are speculating, or test it directly.

This conclusion is based on our finding that the amplitude of sEPSCs is reduced with PLX treatment in developing abGCs (Figure 6D).